# A GPCR-based yeast biosensor for biomedical, biotechnological, and point-of-use cannabinoid determination

Karel Miettinen [1], Nattawat Leelahakorn [1], Aldo Almeida [1,2], Yong Zhao [1], Lukas R. Hansen [1], Iben E. Nikolajsen [1], Jens B. Andersen [3], Michael Givskov [3], Dan Staerk [4], Søren Bak [1] & Sotirios C. Kampranis [1]✉

Eukaryotic cells use G-protein coupled receptors to sense diverse signals, ranging from chemical compounds to light. Here, we exploit the remarkable sensing capacity of G-protein coupled receptors to construct yeast-based biosensors for real-life applications. To establish proof-of-concept, we focus on cannabinoids because of their neuromodulatory and immunomodulatory activities. We construct a $CB_2$ receptor-based biosensor, optimize it to achieve high sensitivity and dynamic range, and prove its effectiveness in three applications of increasing difficulty. First, we screen a compound library to discover agonists and antagonists. Second, we analyze 54 plants to discover a new phytocannabinoid, dugesialactone. Finally, we develop a robust portable device, analyze body-fluid samples, and confidently detect designer drugs like JWH-018. These examples demonstrate the potential of yeast-based biosensors to enable diverse applications that can be implemented by non-specialists. Taking advantage of the extensive sensing repertoire of G-protein coupled receptors, this technology can be extended to detect numerous compounds.

[1] Biochemical Engineering Group, Plant Biochemistry Section, Department of Plant and Environmental Sciences, University of Copenhagen, Thorvaldsensvej 40, 1871 Frederiksberg C, Denmark. [2] Bioremediation Laboratory, Faculty of Biological Sciences, Autonomous University of Coahuila, Carretera Torreón-Matamoros km. 7.5, Torreón, Coahuila 27000, Mexico. [3] Costerton Biofilm Center, Department of Immunology and Microbiology, Faculty of Health and Medical Sciences, University of Copenhagen, Blegdamsvej 3B, 2200 Copenhagen, Denmark. [4] Department of Drug Design and Pharmacology, Faculty of Health and Medical Sciences, University of Copenhagen, Universitetsparken 2, 2100 Copenhagen, Denmark. ✉email: soka@plen.ku.dk

G-protein-coupled receptors (GPCRs) are the main sensing entities of higher eukaryotes. They confer the ability to see, smell, and taste, and play key roles in endocrine signaling and the regulation of the immune system[1]. As such, GPCRs have evolved to detect molecules with tremendous chemical diversity, from peptides to small chemical compounds and even light. Harnessing the sensing capability of GPCRs could have profound implications for biotechnology, enabling specific detection of an immense diversity of ligands. Whole-cell biosensors based on microbial cells, such as *Saccharomyces cerevisiae*, armed with GPCRs as sensing entities could provide robust detection of different molecules with considerable advantages in terms of cost, diversity and portability. Work in this direction has established a general framework for integrating heterologous GPCR signaling in yeast[2–5]. However, achieving the sensitivity, throughput, and ease-of-use that is essential for most biotechnological applications requires further development. In this work, we choose a key application, the determination of cannabinoid compounds, to further develop this technology and showcase its performance in challenging, real-life, problems.

Cannabinoids, the bioactive compounds of cannabis plants, have potent analgesic and anti-inflammatory properties and have been used in traditional medicine for millennia[6]. However, in the early 20th century, cannabis was made illegal due to its psychoactive effects, and, as a result, cannabinoids have largely been neglected by modern medicine. Recently, a strong interest in cannabinoids has re-emerged as several studies have demonstrated that cannabinoids have the potential to delay the progression of neurodegenerative diseases such as Alzheimer's, Huntington's, and multiple sclerosis[7,8]. This interest is reflected in more than 500 currently ongoing clinical trials involving cannabis or cannabinoids worldwide (clinicaltrials.gov)[9]. In combination with the decriminalization of the use of cannabis for recreational and medicinal use, the demand for cannabis and its products has sharply risen and the world market for cannabinoid-based pharmaceuticals is expected to reach $25 billion by 2025[10].

Cannabinoids exert their activity in humans by targeting the canonical cannabinoid receptors $CB_1$[11] and $CB_2$[12]. In addition to (-)-*trans*-$\Delta^9$-tetrahydrocannabinol (THC), cannabidiol (CBD), and the other structurally related molecules found in cannabis plants, the cannabinoid receptors are also targeted by several structurally non-related types of compounds. This diverse group of ligands encompasses endocannabinoids (the endocrine signaling molecules naturally synthesized by humans and other animals[13,14]), other structurally-distinct plant natural products (termed collectively as phytocannabinoids[15]), and different types of synthetic compounds (many of which are used as illicit drugs)[16,17]. For simplicity, herein, we use the term cannabinoids to collectively refer to any ligand of $CB_1$ and $CB_2$, regardless of structure or origin.

Both $CB_1$ and $CB_2$ are GPCRs. $CB_1$ is present in the central nervous system and its main role is to modulate neurotransmitter release[14]. It is the main receptor responsible for the psychotropic effect of cannabinoids. In contrast, $CB_2$ is found in cells of the immune system[18–22] and is involved in inflammation. Since $CB_2$ activation is regarded as devoid of psychotropic effects, it is considered a potential therapeutic target for the treatment of conditions where very limited therapies exist, such as Alzheimer's, multiple sclerosis or arthritis. The rapidly rising interest in cannabinoids as therapeutics has resulted in a sharp increase in the demand for canonical cannabinoids but also for natural or synthetic compounds specifically targeting $CB_2$.

Currently, compounds acting on $CB_2$ are evaluated using different mammalian cell-based assays. These require dedicated equipment and personnel, keeping the characterization of new chemical compounds out of the reach of many chemical biology or natural product chemistry labs. Thus, developing cost-effective and easy-to-use tools for identifying GPCR ligands could speed up drug discovery by enabling more academic and small-size commercial labs to test compound libraries or natural extracts. Screening of complex plant or microbial extracts, in particular, requires assays with high sensitivity and selectivity, as these compounds are often present in minute amounts in highly complex mixtures (Fig. 1a). Moreover, in cell-based assays, the different cannabinoid receptor ligands frequently display off-target effects (e.g., on GPR55, COX-2, or transient receptor potential (TRP)-channels[23,24]) requiring validation of their specificity. Efficient orthogonal assays in non-mammalian systems can help addressing this limitation.

In addition, the decriminalization of the use of cannabis for self-medication and recreation in several countries creates a need for facile methods to determine the potency of commercial cannabis preparations. Moreover, there is an urgent need to detect new designer drugs (synthetic cannabinoids) that regularly emerge on the market and have been associated with severe cases of poisoning[17,25,26]. As these are new drugs, there is typically no go-to detection method available. In addition, there is a considerable increase in the number of supervised therapeutic approaches that employ cannabinoids for the treatment of chronic and inflammatory pain[27], which requires regular monitoring of levels of cannabinoids and their metabolic products in body fluids during the course of treatment. Many of these challenges require the development of convenient and robust methods so that non-trained personnel outside analytical laboratory settings (e.g., in point-of-care facilities or medical practitioners' premises) can detect the presence of cannabinoids in real-life samples such as those obtained from urine or saliva (Fig. 1a).

The broad chemical diversity of compounds that modulate the cannabinoid receptors poses a considerable challenge on the development of a single method that can efficiently and reliably address all the above needs. Here, we propose that the solution to this is a yeast whole-cell biosensor that uses the $CB_2$ receptor to detect the whole range of its structurally diverse ligands. Baker's yeast has been shown to be suitable for the functional expression of several GPCRs, including the $CB_1$ and $CB_2$ receptors[2,4,5]. This is possible because GPCR signaling pathways, such as the yeast pheromone pathway, share a highly conserved architecture consisting of analogous components between kingdoms. As a general mechanism, upon ligand binding, each GPCR receptor activates a dedicated Gα protein that, in turn, dissociates from the heterotrimeric Gαβγ complex. In the yeast pheromone pathway, the resulting Gβγ dimer triggers a MAPK cascade, which in turn activates a transcription factor (Ste12p) that finally drives the expression of pheromone response genes. In previous work, it has been possible to hijack the yeast pheromone pathway by replacing the pheromone receptor with a GPCR of interest and monitor the pathway's downstream response using a reporter gene[4,5,28–30]. This has been broadly exploited for example, in the study of specific receptors[31,32], deorphanization of uncharacterized receptors[29], study of cell–cell communication[33], guiding metabolic engineering efforts[34], and detection of fungal pathogens[28]. However, specialized yeast biosensors capable of performing low-cost high-throughput bioactivity characterization, bioprospecting, and, especially, out-of-lab applications, have yet to be introduced.

In this work, we develop a flexible modular cannabinoid biosensor by coupling the human $CB_2$ receptor to the yeast pheromone-signaling pathway (Fig. 1b). We optimize the developed biosensor to achieve similar sensing dynamics than mammalian cell-based systems[24] in a far more economical and user-friendly format. We further expand the performance of the system by developing dedicated color- and luminescence-based reporter strains to meet the specific requirements of different

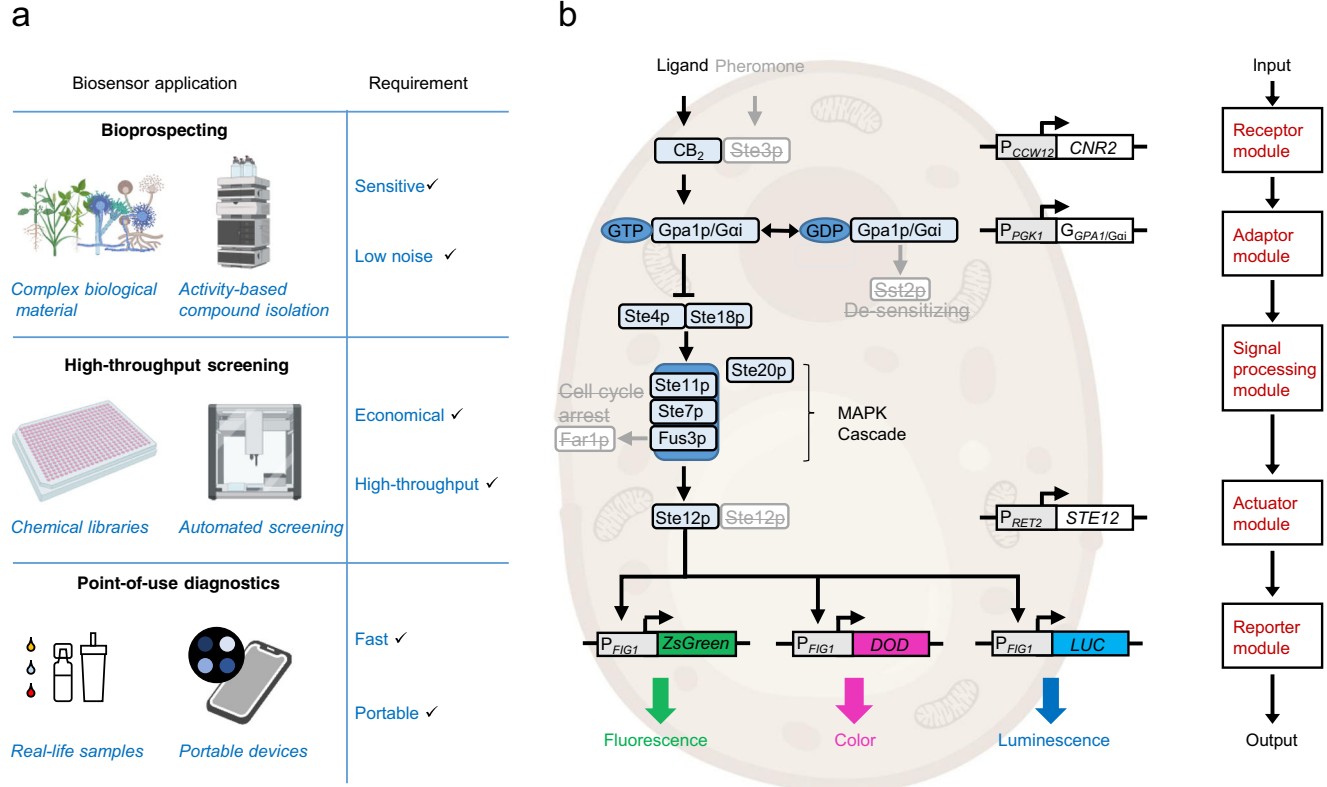

**Fig. 1 The CB$_2$ cannabinoid biosensor design. a** The biosensor was developed to enable diverse applications with different requirements. For example, bioprospecting of complex biological material requires the biosensor to be sensitive but with low background. This is because the bioactive compounds are often present in minute amounts among many other compounds potentially interfering with detection. On the other hand, screening of chemical libraries requires a biosensor that is robust, economical, and amenable to high-throughput workflow. For this, fast-growing and easy-to-prepare cells that can be handled with non-expensive material and equipment are desirable. In the case of a biosensor for point-of-use diagnostics outside the lab, this needs to be easy to use, fast, and operable by the equipment available to non-experts. **b** The cannabinoid biosensor is based on a modular design. Interchangeable parts can be introduced into the receptor, adaptor, actuator, or reporter modules (red), while the native yeast Gβ and Gγ subunits and the MAPK cascade are employed as a signal-processing module without further modification. The parts are integrated into a chassis strain KM111 where genes encoding the yeast pheromone pathway components to be replaced (pheromone receptor *STE3*, Gα subunit *GPA1*, and pheromone pathway master regulator *STE12*) have been removed (strikethrough) alongside with *SST2* (which returns Gα to its inactive state) and *FAR1* (which triggers cell-cycle arrest). This design enables the functional insertion of different GPCR receptors by pairing them with the corresponding Gpa1p/Gα chimera. According to the specific requirements of each application, the biosensor can be fitted with an optimal reporter construct including, for example, a fluorescence, color, or luminescence reporter.

demanding applications and showcase the biosensor's performance in three case studies. To demonstrate the biosensor's high-throughput screening capacity, we present here the discovery of two CB$_2$ agonists and two CB$_2$ antagonists from a compound library of 1600 synthetic compounds. To showcase the sensor's ability to cope with highly complex biological samples, we apply it in the bioprospecting of 71 extracts derived from different parts of 54 different medicinal plants and describe the bioactivity-guided isolation of a agonist of CB$_2$, dugesialactone. Finally, we demonstrate the use of a biosensor as a sensitive portable device for detecting cannabinoids from reconstructed saliva samples.

Our results demonstrate that the extensive sensing repertoire of GPCRs can be harnessed to establish robust whole-cell biosensors. This technology can now be extended to detect numerous other molecules, from small compounds to proteins, enabling advanced biotechnological applications.

## Results

**Constructing the chassis for the GPCR-based biosensor.** The GPCR signaling mechanism is inherently modular and can be abstracted in the form of five linearly connected modules (Fig. 1b). The input module comprising the GPCR protein, the

adaptor module that contains the dedicated Gα protein, the signal-processing module that encompasses the MAPK cascade, the actuator module that contains the MAPK-controlled transcription factor, and, finally, the output module that includes the activated genes. Our design takes advantage of this modular structure to construct different biosensors by using a basic chassis (or platform) strain and integrating different combinations of parts in the above-mentioned modules (Fig. 1b). This enables flexible setup and functional optimization of the biosensor for different applications by shuffling different component-encoding genes and promoters in each module to find the best-performing configuration.

To enable this modular design, we first constructed the chassis strain by removing the genes encoding for pheromone pathway components that are to be replaced by custom parts (Supplementary Table 1) (Supplementary Fig. 1). Thus, we knocked out the genes for the a-pheromone receptor (*STE3*; to be replaced by GPCR receptor-encoding gene), the Gα subunit (*GPA1*; to be replaced by a Gα gene that is compatible with the chosen GPCR), and the pheromone response master regulator transcription factor (*STE12*; a prerequisite for removing *GPA1*). In addition, we removed two genes that are detrimental to biosensor function (*SST2* and *FAR1*). *SST2* encodes a protein that contributes to

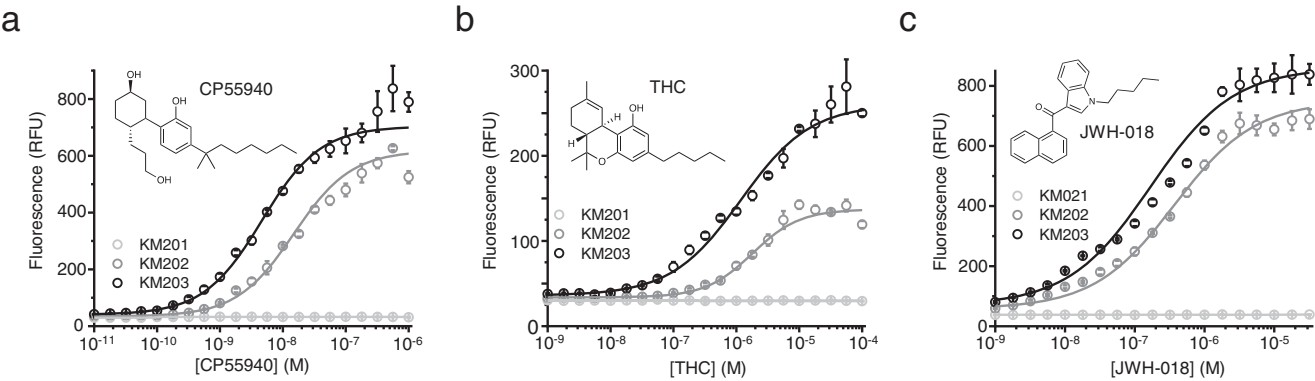

**Fig. 2 Detection of cannabinoids with the fluorescent output cannabinoid biosensor. a** Incubation of the cannabinoid biosensor strain in the absence of CB$_2$ receptor (KM201, light gray) with up to 1 μM CP55940 resulted in no increase of the fluorescent output. Inducing biosensor strain KM202 expressing CB$_2$ receptor (dark gray) with 10 pM to 1 μM CP55940 resulted in a typical sigmoidal dose–response curve revealing an apparent EC$_{50}$ of 15 nM. Strain KM203 (black) producing CB$_2$ fused with the mating factor α signal sequence (MFαSS-CB$_2$) showed higher sensitivity (EC$_{50}$ = 4.6 nM). **b** When the above-mentioned strains were incubated with THC, KM203 (black) showed higher sensitivity (EC$_{50}$ = 1.2 μM) than KM202 (dark gray), which displayed an apparent EC$_{50}$ of 1.8 μM. The control strain, KM201 (light gray), was not induced by the cannabinoid. **c** The biosensor was also incubated with JWH-18. For KM202, EC$_{50}$ was determined to be 370 nM, while for KM203, EC$_{50}$ was 169 nM. The control, KM201 was not induced by JWH-018. Data presented as mean +/− standard deviation. $n$ = 3 biologically independent samples. Source data are provided in the Source Data file.

returning Gα to its non-activated state[35], resulting in attenuated signaling through the pathway, and *FAR1* causes yeast to enter cell-cycle arrest following pheromone sensing. The strain AM254[36] was used as parent as it was predicted to be advantageous for producing plasma membrane-localized proteins, such as GPCRs, which translocate via the endoplasmic reticulum. This resulted in the chassis strain KM111, which serves as a basis for all subsequent biosensor strains.

**Constructing the initial cannabinoid biosensor**. The initial cannabinoid biosensor strain KM202 (Supplementary Fig. 2) was constructed by integrating four genes (Supplementary Fig. 3) into the X-3 locus[37] (known to be well suited for heterologous gene expression[38] of chassis strain KM111 (Supplementary Table 1). These include the human cannabinoid receptor CB$_2$ gene (*CNR2*), a hybrid gene encoding for a chimera between the yeast Gα protein and the five C-terminal amino acids of human Gαi1 (*GPA1/Gai1*) capable of linking the receptor to the downstream pathway[2,3], the pheromone pathway master regulator (*STE12*), and the fluorescent reporter (ZsGREEN[39]). To correctly balance the pathway components, the above-mentioned genes were put under the control of specific promoters, according to the findings of Shaw and co-workers[5]. Thus, *CNR2* (fused with a sequence encoding for three copies of the hemagglutinin (HA)-tag to facilitate monitoring of expression) was put under the control of the strong constitutive promoter P$_{CCW12}$, *GPA1/Gai1* under the medium/strong constitutive promoter P$_{PGK1}$, *STE12* under the medium strength constitutive promoter P$_{RET2}$[40] and the reporter gene ZsGREEN under the native pheromone response promoter P$_{FIG1}$ that is activated by Ste12p[41]. To evaluate the performance of the initial biosensor, it was tested with a concentration series of the potent CB$_2$ full agonist CP55940. This resulted in a typically-shaped dose–response curve (Fig. 2a). The calculated EC$_{50}$ was 15 ± 1.53 nM, which is at the high end of the range of mammalian cell-based CB$_2$ assays (typically 3–11 nM CP55940)[24]. The limit of detection (LOD) was determined to be 56 pM and the maximum signal-to-noise ratio (SNR) was 17.7:1.

We then tested the usefulness of the biosensor in the detection of compounds present in different cannabis products, such as the principal psychoactive constituent of cannabis (-)-*trans*-Δ$^9$-tetrahydrocannabinol (THC) and the synthetic cannabinoid JWH-018. JWH-018 is present in Illegal synthetic cannabinoid

preparations, like Spice or K2, found on the street market[42]. Like many other designer drugs, JWH-018 cannot be detected with available (antibody-based) cannabinoid quick kits. The biosensor was able to detect both compounds. The corresponding dose–response curve for THC (Fig. 2b) revealed values for EC$_{50}$ of 1.8 ± 0.24 μM, SNR of 4.2:1, and LOD of 100 nM, while JWH-018 was detected with EC$_{50}$ of 370 ± 25 nM, SNR of 16.5, and LOD of 1 nM (Fig. 2c).

**Optimization of the basic biosensor**. Biosensor sensitivity, evaluated here by the EC$_{50}$ and LOD values, has been shown to be directly related to the affinity of the receptor for the ligand and the number of active receptors on the cell surface[5,35]. Previous work has demonstrated that the degree of membrane localization is an important bottleneck for heterologous GPCR functionality and differs greatly between heterologously expressed receptors[43,44]. Appending the yeast mating factor α (prepro) secretion signal (MFαSS) to the N-terminus of the receptor has been shown to enhance both the total expression and membrane localization of GPCRs in yeast[44]. Thus, in order to enhance the sensitivity and overall output of the cannabinoid biosensor, we constructed biosensor strain KM203, where CB$_2$ is fused to MFαSS and expressed from the strong yeast promoter P$_{CCW12}$. This resulted in improved plasma membrane localization of the CB$_2$ receptor and increased total levels of the receptor protein (Supplementary Figs. 4 and 5). When the biosensor (strain KM203) was tested with CP55940 and its dose–response curve was compared with that of strain KM202 (no MFαSS), a clear improvement in biosensor sensitivity and output level was observed (Fig. 2a). In the case of CP55940, the EC$_{50}$ improved 3.2 times to 4.64 ± 0.44 nM, the LOD decreased 5.6 times to 10 pM, and the maximum SNR improved to 20.1:1. The determined EC$_{50}$ now reflects closely the equilibrium binding constant for CP55940 (determined to be 3.6 nM)[24], suggesting that in the optimized strain (KM203) there is the efficient coupling of the binding event to the reporter signal. Overall, the observed EC$_{50}$ for CP55940 is on a par with that of most mammalian cell-based signaling assays (e.g., 3.47 nM in the GTPγS assay, 4.07 nM in the β-AR recruitment assay)[24]. When the optimized biosensor (KM203) was evaluated for its ability to sense THC (Fig. 2b), the EC$_{50}$ improved to 1.2 μM (1.5-fold), LOD to 32 nM (3.1-fold), and SNR to 7.4:1 (1.8-fold), compared to KM202 (Fig. 2c).

KM203 also exhibited improved detection of JWH-018 with EC50 of $169 \pm 27$ nM (2.2-fold), SNR of 15:1, and LOD of 1 nM. Finally, we determined the optimal temperature and assay time for KM203. At saturation, the biosensor showed the highest output at 25 °C (Supplementary Fig. 6). However, the fastest response was observed at 30 °C, where $T_{50}$ (the time to reach 50% of maximum output) was 4.6 h (Supplementary Fig. 7).

**Betalain-based reporters for portable biosensors.** To further improve the applicability of the developed biosensor, we focused on optimizing the performance of the output module. The current biosensor utilizes the fluorescent protein ZsGREEN, which provides a relatively direct relationship between reporter expression and signal magnitude (number of fluorescent molecules) but is confined to measurements in a laboratory setting. To enable analysis in real-life settings, the performance of the output module can be enhanced in two ways. First, by switching from fluorescence to an output that can be detected without a dedicated instrument, such as color or light, which will enable portable applications. Second, by using an enzyme-based reporter instead of fluorescence, which will add a signal amplification step, thus enabling detection with low-gain devices such as mobile phones.

Typical colorimetric reporters used in yeast include glycosidases[45] and carotenoid biosynthesis genes[28]. However, these have important drawbacks that include slow color build-up and the requirement for cell disruption. Therefore, we set out to develop a new faster and non-cell-disruptive color reporter. For this, we focused on betalains, the intense pigments found in red beet (*Beta vulgaris)* and constructed three reporter systems producing different colors.

The group of betalains includes two categories of compounds, the purple-red betacyanins and the orange betaxanthins (Fig. 3a). To produce the betacyanin betanin in yeast, we introduced the tyrosine hydroxylase/cyclase from *B. vulgaris* (*Bv*CYP76AD1), the DOPA dioxygenase from *Mirabilis jalapa* (*Mj*DOD), and the DOPA-5-glucosyltransferase from *B. vulgaris* (*Bv*DOPA5GT)[46] to construct biosensor strain KM204 (Fig. 3a and Supplementary Table S1). In this strain, *Mj*DOD serves as the effective reporter gene controlled by the pheromone responsive $P_{FIG1}$ promoter, whereas *Bv*CYP76AD1 and *Bv*DOPA5GT were constitutively expressed. When KM204 was grown on agar plates containing 1 µM CP55940, red-colored colonies were clearly visible suggesting that the biosensor was functional (Supplementary Fig. 8). However, color build-up in liquid cultures of KM204 was very slow, suggesting that although using betanin as reporter could be useful in library screening experiments, a faster responding reported would be preferable for other applications such as, for example, a portable biosensor. Thus, we turned to the yellow betaxanthins and constructed a biosensor strain (KM205) co-expressing *Mj*DOD controlled by the $P_{FIG1}$ promoter together with the tyrosine hydroxylase *Bv*CYP76AD5 from *B. vulgaris* (Fig. 3a), as this combination has been shown to produce betaxanthins in yeast[46]. This time, in addition to obtaining yellow-colored colonies on CP55940-containing agar plates (Supplementary Fig. 8), measuring the absorbance of the KM205 culture supernatant showed a dose-dependent biosensor signal (Supplementary Fig. 9).

Encouraged by this finding, we continued to develop a biosensor with even faster and stronger color output. The betalain precursor betalamic acid can readily react with several primary and secondary amines to produce compounds with intense colors[46]. Thus, to develop a reporter with strong red color, we supplemented cultures of KM205 with 0.5 mM O-dianisidine (O-da) to obtain O-da-betacyanin. Furthermore,

because it has been shown that betalain production can be improved by adding tyrosine in the yeast media, we also added either tyrosine or L-DOPA (the next step in the pathway) to boost color production. A clearly enhanced color signal was obtained (Fig. 3b). As KM205 with O-da and L-DOPA showed the most distinguishable color, we chose to use this combination in the rest of this work and refer to it as the betalain reporter. Testing the betalain reporter strain with a dilution series of CP55940 revealed that the color signal was detectable by the eye down to 100 pM (Fig. 3c). Quantification of the color signal from this series by measuring absorbance at 520 nm, the O-da-betacyanin peak wavelength (Supplementary Fig. 10), showed an apparent EC50 of $7.05 \pm 1.62$ nM, LOD of 63 pM and SNR 26.3 (Fig. 3d), indicating comparable sensitivity with the fluorimetric strain KM203. The colorimetric biosensor could also detect THC (Fig. 3e; $EC_{50} = 2.05 \pm 0.11$ µM, LOD = 32 nM, SNR = 11.3) and JWH-018 (Fig. 3f; $EC_{50} = 221 \pm 11$ nM, LOD = 3.2 nM and SNR = 20.7:1) with comparable performance as KM203. For KM205, the optimal temperature for the maximum (saturated) output was 25 °C (Supplementary Fig. 6), but, as for KM203, incubation at 30 °C resulted in the fastest response ($T_{50} = 6.1$ h) (Supplementary Fig. 7).

**A luminescence reporter improves dynamic range and speed.** Although our developed fluorescence and betalain reporter strains are already well suited for cannabinoid detection inside and outside the lab, some key applications such as the determination of cannabinoids in complex biological material present additional challenges, as many biological samples are likely to contain colored or fluorescent molecules. Similarly, yeast cells, under specific conditions may display high levels of auto-fluorescence or produce small amounts of colored molecules. Thus, developing a biosensor with a reporter less prone to background noise would be beneficial. In addition, shortening the response time for the biosensor will be desirable for certain applications. We set out to develop a biosensor strain to address both these requirements.

We opted for a luminescence-based reporter because its output is light, which avoids background or interference from the sample or the chassis. To achieve a strong luminescence output, we chose the NanoLuc luciferase, as it has been shown to produce the brightest signal among similar enzymes[47]. We constructed the luminescence reporter strain KM206 (Supplementary Table 1) by placing NanoLuc expression under the control of $P_{FIG1}$ in the reporter module. When tested with CP55940, the biosensor produced sufficient output to be measured with a cellular phone camera (Fig. 4a). Following optimization of the analysis parameters (Supplementary Figs. 11 and 12), we evaluated the sensitivity of the biosensor. A dose–response curve with CP55940 (Fig. 4b) showed an apparent $EC_{50}$ of $1.39 \pm 0.07$ nM, while exhibiting minimal noise (0.13% of the maximum signal). The LOD was determined to be 10 pM and the SNR 687:1. Moreover, dose-repose curves from three separate experiments showed a day-to-day drift of 5.6% (standard deviation) in $EC_{50}$ (Supplementary Fig. 13). To ensure consistency in applications where high resolution is needed, we also verified that cannabinoid ligands stay in solution rather than being absorbed into the cell membrane (Supplementary Fig. 14).

As the temperature is known to affect the affinity of a GPCR for its ligands ($K_D$), we evaluated the change in sensitivity of KM206 for CP55940 between different temperatures (Supplementary Fig. 15). In the range from 20 °C to 37 °C, an average change of 7% in $EC_{50}$ per °C was observed. As with the other biosensor strains developed here, the maximum output was reached at 25 °C (Supplementary Fig. 6). For KM206, however,

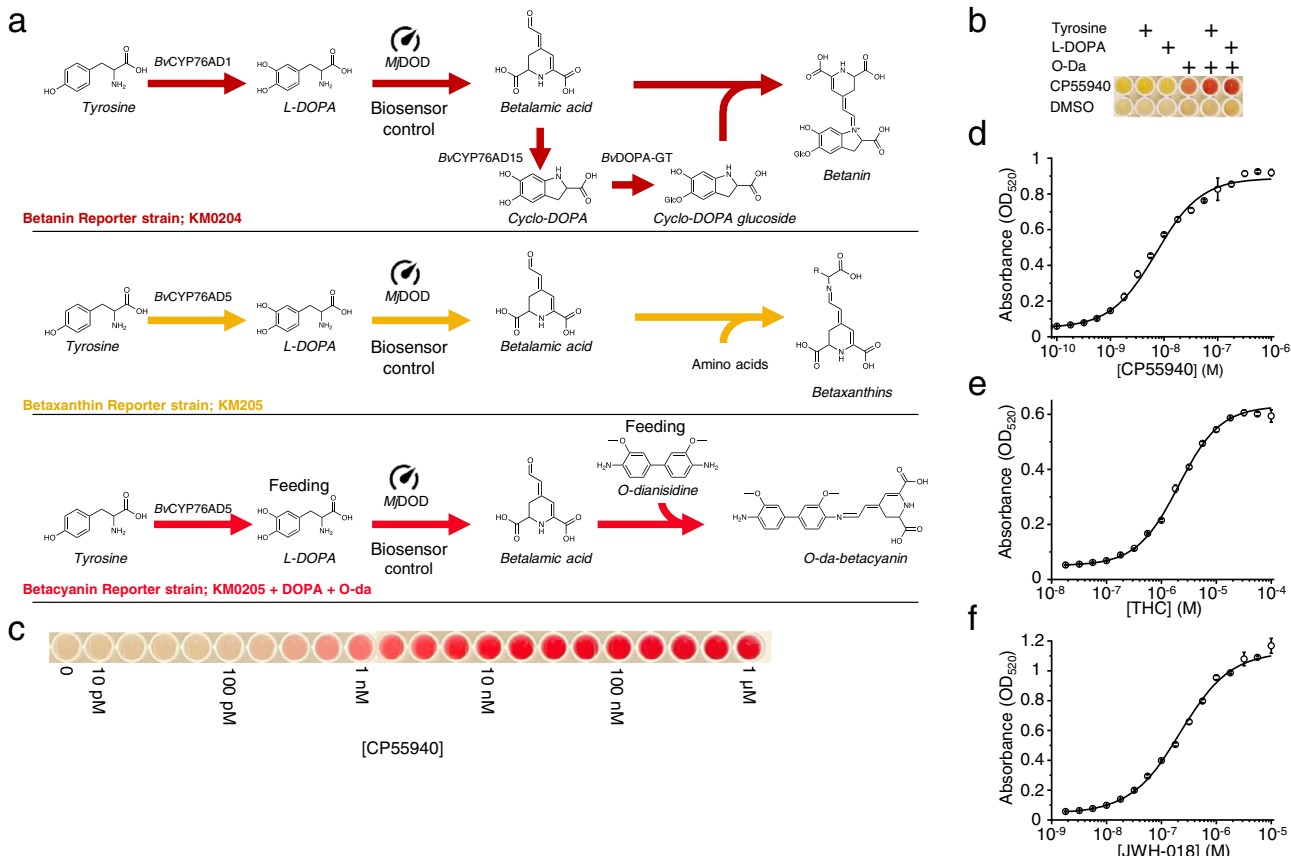

**Fig. 3 Betalain biosynthesis-based reporters. a** Three betalain-based transcriptional reporter systems were constructed based on three different biosynthetic pathways. In all the reporters, DOPA dioxygenase (*Mj*DOD) is the reporter gene that is controlled by the biosensor and other genes are constitutively expressed. The betanin reporter is based on the strain KM204, where *Bv*CYP76AD1 hydroxylates tyrosine into L-DOPA that is subsequently oxidized to betalamic acid. This is then further oxidized by *Bv*CYP6AD1 into cyclo-DOPA and glucosylated by DOPA-5-glucosyltransferase to cyclo-DOPA-glucoside. Finally, betalamic acid and cyclo-DOPA-glucoside spontaneously condense to make betanin. The betaxanthin reporter is based on strain KM205, where CYP76AD5 is used to hydroxylate tyrosine into L-DOPA, which is then further oxidized into betalamic acid. Betalamic acid will spontaneously react with amino acids available in the cells to make betaxanthins. The O-dianisidine-betacyanin reporter is also based on strain KM205. In this case, however, L-DOPA is directly fed to the cells and oxidized by *Mj*DOD to betalamic acid, which subsequently reacts strongly with O-dianisidine to produce O-da-betacyanin. **b** The addition of supplements to strain KM205 modulates the color output. The addition of O-da results in the appearance of a red color. Color intensity is strongly increased upon supplementation with both O-da and L-DOPA. **c** The color produced by the O-da-betacyanin reporter can be easily detected by eye. Inducing the strain KM205 with a dilution series ranging from 10 pM to 1 μM CP55940 for 16 h and supplementing it with L-DOPA and O-da results in different intensities of red, where the effect of 100 pM CP55940 could still be detected by eye. **d–f** The output of the O-da-betacyanin reporter can be quantified by measuring the absorbance of the pigment at 520 nm. **d** A dose–response curve of the betacyanin reporter strain induced with CP55940 shows an apparent $EC_{50}$ of 7 nM. **e** For THC, the dose–response curve reveals an apparent $EC_{50}$ of 2 μM. **f** Dose response for JWH-018 shows an apparent $EC_{50}$ of 221 nM. For (**d–f**), data presented as mean +/− standard deviation. $n = 3$ biologically independent samples. Source data are provided in the Source Data file.

the fastest response time was also at 25 °C ($T_{50} = 1.5$ h). Because of the improved resolution of this biosensor, a signal strong enough for the measurement of cannabinoids (SNR > 25:1) could be observed already at 30 min (Supplementary Fig. 7), suggesting that KM206 can be used for applications that require fast response. Furthermore, the response time was found to be very similar between different ligands tested (Supplementary Fig. 16).

To further improve the system, we developed a null control strain for detecting off-target (non-CB$_2$-specific) activities of potential ligands including, for example, compounds toxic to the yeast or allosteric modulators of biosensor components. This strain, KM207, was constructed by replacing CB$_2$ in KM206 with the A$_{2A}$ adenosine receptor[48] (Supplementary Fig. 17). By including KM207 in the biosensor workflow, we were able to evaluate whether the tested compounds interfered (negatively or positively) with the function of any other module of the biosensor except for the GPCR. KM207 was used to produce dose–response

curves (in the presence of a fixed concentration of adenosine) for each compound studied in this work, which served as control curves (shown in gray in Fig. 4b–k).

To assess the luminescence biosensor's performance, we produced dose–response curves for THC, the synthetic cannabinoid JWH-018, and the liver metabolite of THC, 11-OH-THC, which is commonly found in samples from patients treated with THC. The biosensor was able to detect THC with $EC_{50} = 5.18 \pm 0.98$ μM, LOD of 100 nM, SNR of 283:1 and JWH-018 with $EC_{50}$ 176 ± 24 nM, LOD 10 nM, SNR 50.2:1 (Fig. 4c, d). Moreover, the biosensor was able to reveal the presence of 11-OH-THC with $EC_{50} = 3.11 \pm 0.28$ μM, LOD 100 nM, and SNR 30.1:1, despite the inhibitory off-target effect of 11-OH-THC on the assay, revealed by the null control (Fig. 4e). Judging from the corresponding null control curves, the other compounds did not show off-target effects. This shows that this biosensor can be applied in the determination of illicit drugs

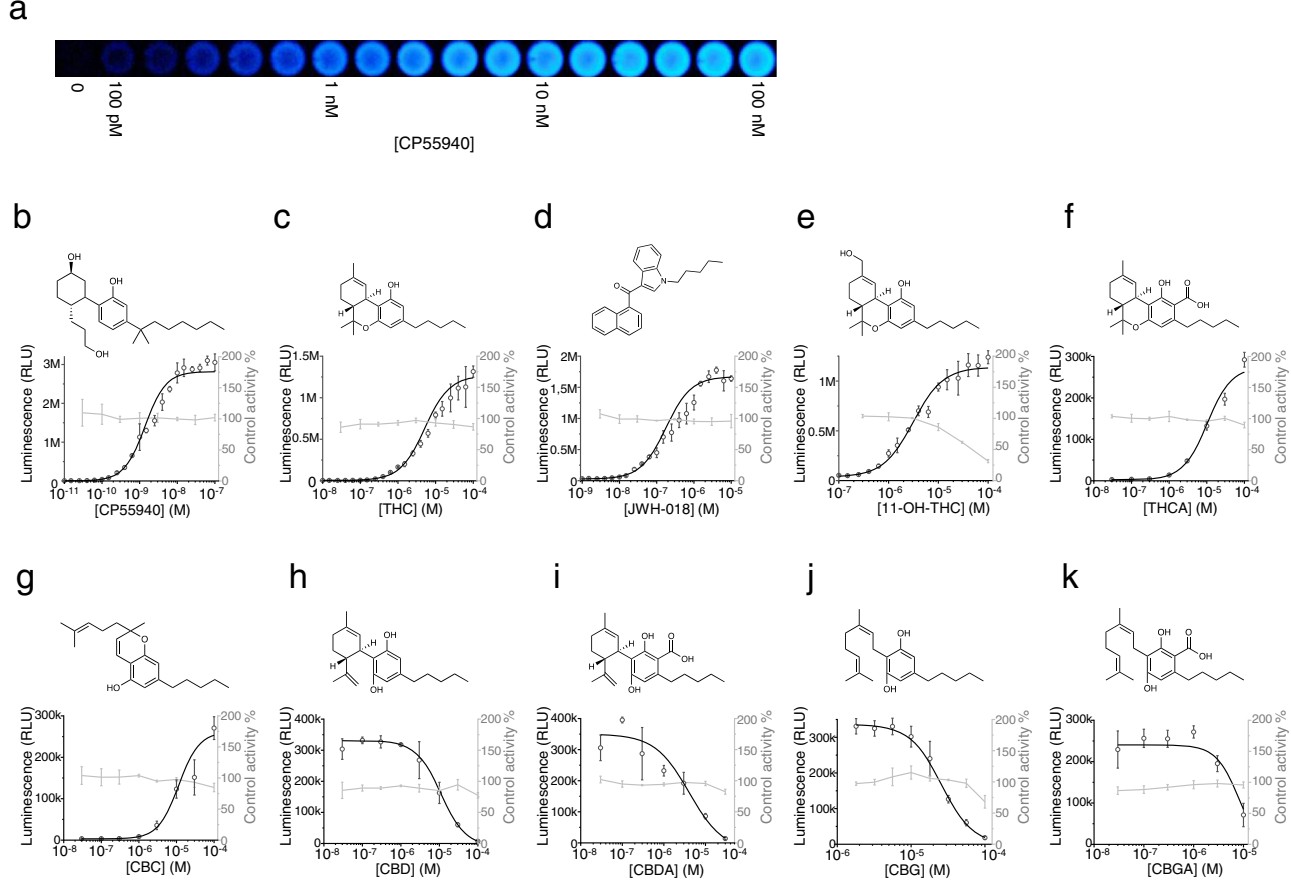

**Fig. 4 A cannabinoid biosensor with a luciferase reporter. a** The biosensor strain KM206 employs a luciferase reporter. It was incubated with a dilution series of CP55940 ranging from 100 pM to 100 nM and photographed it with a cell phone camera. This resulted in a clearly visible signal until 100 pM CP55940. **b–k** The response of the strain KM206 to different cannabinoids (black) was determined by incubating with a dilution series of each ligand and the resulting luminescent signal was measured using a plate reader. Control curves (gray) depict the effect of the corresponding concentrations of each ligand on the control strain KM207 activated with 1 mM adenosine. **b** Dose response of the strong synthetic cannabinoid CP55940. **c** Dose–response curve with THC, the main cannabinoid from cannabis. **d** Dose–response curve with 11-OH-THC, the liver metabolite of THC. **e** Dose–response curve of the structurally unrelated strong synthetic cannabinoid JWH-018. **f–k** Dose–response curves of the cannabis constituents (-)-trans-$\Delta^9$-tetrahydrocannabinolic acid (THCA), cannabichromene (CBC), cannabidiol (CBD), cannabidiolic acid (CBDA), cannabigerol (CBG) and cannabigerolic acid (CBGA) respectively. Data presented as mean +/− standard deviation. $n = 3$ biologically independent samples. Source data are provided in the Source Data file.

and in the monitoring of cannabinoid clearance in samples from patients or users. In addition, we assessed the effect of several natural cannabinoids that could be relevant for the analysis of cannabis preparations (Fig. 4f–k). Cannabichromene (CBC) and (-)-trans-$\Delta^9$-tetrahydrocannabinolic acid (THCA) were found to be medium-low potency activators of the biosensor ($EC_{50}^{CBC} = 11.4 \pm 2.84\,\mu M$ and $EC_{50}^{THCA}\ 10.02 \pm 2.13\,\mu M$), while cannabidiol (CBD), cannabidiolic acid (CBDA), cannabigerol (CBG) and cannabigerolic acid (CBGA) were found to be medium-low potency inhibitors of the biosensor ($IC_{50}^{CBD} = 10.84 \pm 0.73\,\mu M$, $IC_{50}^{CBDA} = 4.05 \pm 2.17\,\mu M$, $IC_{50}^{CBG} = 25.75 \pm 1.41\,\mu M$, $IC_{50}^{CBGA} = 8.36 \pm 0.87\,\mu M$).

**Case study 1: discovery of CB₂ agonists and antagonists.** Having, established cannabinoid biosensor strains optimized for sensitivity and containing fluorimetric, colorimetric, and luminometric output modules, we proceeded to testing them in three case studies with increased difficulty. Currently, most new GPCR-targeting drugs are discovered by high-throughput screening of synthetic chemical libraries using mammalian cell cultures. Such experiments are costly and require highly specialized equipment and infrastructure and are, therefore, typically limited to specialist

labs or pharmaceutical companies. However, we argue that using a microbial whole-cell biosensor for the initial stages of screening and the shortlisting of candidates for further mammalian cell-based or animal studies could help lowering the cost of such experiments. Moreover, the development of simple and economical biosensor strains that can freely be co-developed and exchanged between researchers would make compound screening available to non-specialist or lower-budget research labs that have access to publicly available or in house-constructed compound libraries. Thus, we started developing a yeast-based screening platform based on our CB₂ cannabinoid biosensor. Aiming to devise a solution for non-specialist, limited-budget labs, we based our method on the inexpensive, open-source, liquid handling robot Opentrons OT-2 and a common plate reader (Molecular devices M5) in 384-well plate format (Fig. 5a).

As proof-of-concept, we screened 1600 randomly chosen compounds from the total >55,000 compounds present in the chemical compound collection of the Chemical biology and HTS facility of the University of Copenhagen (https://cbhts.ku.dk/chemical-compound-collection/). We started developing the high-throughput screening method by performing an agonist screen using the betalain biosensor strain (KM205) as it directly displays color signal and does not require cell lysis or luciferin

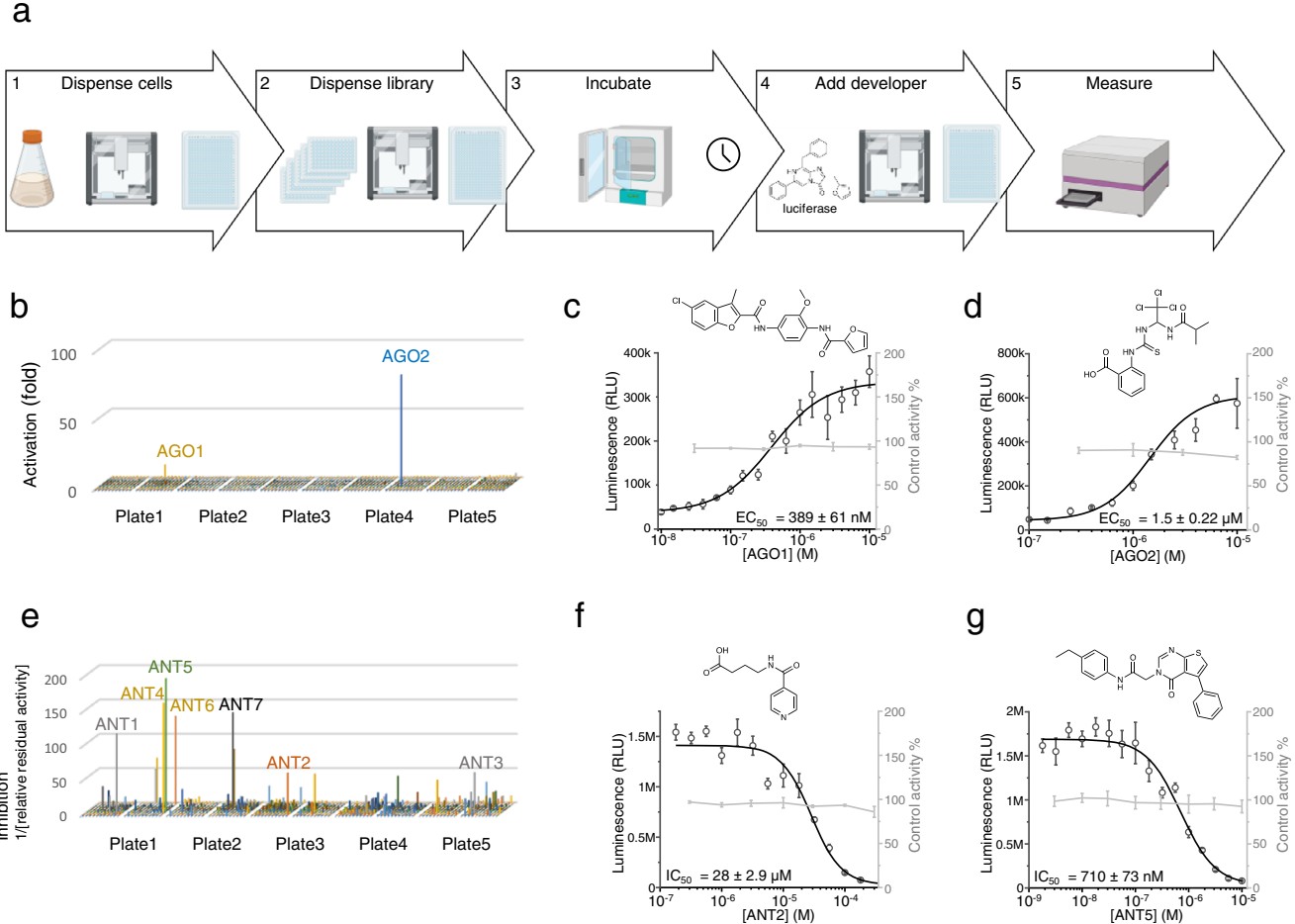

**Fig. 5 High-throughput screening applications. a** The high resolution and low background of the KM206 cannabinoid biosensor strain makes it well suited for high-throughput screening for GPCR ligands. The HTS screening workflow consists of (1) robotically dispensing the biosensor strain into 384-well plates, followed by (2) dispensing the library into these plates, (3) incubating for 3 h, (4) adding the developer solution (lysis buffer with luciferin) to the cells, and (5) measuring luciferase activity with a plate reader. **b** An agonist screen of the library led to the discovery of 2 $CB_2$ agonists with markedly higher luciferase signal than the background. **c** A Dose–response curve (black) for AGO1 revealed an $EC_{50}$ of 200 nM. While control curve (gray) shows no noticeable non-$CB_2$ specific effect of AGO1 on the control strain KM207. **d** In the case of AGO2, a dose–response curve showed an $EC_{50}$ 2 of μM and no effect on control strain. These results clearly validate the hits from the screening experiments. **e** An antagonist screen of the library was performed by adding the library on top of cell supplemented with 2 nM CP55940. Here, compounds that resulted in lower luciferase activity than that of CP55940 alone were defined as hits (inhibitors). Their potency was calculated as [relative residual activity]$^{-1}$. This revealed several potential antagonists (ANT1-7). **f** A dose–response curve for ANT2 revealed an $IC_{50}$ of 29 μM. The control curve (gray) shows no non-$CB_2$ specific effect of ANT2 on the control strain KM207. **g** For ANT5, a dose–response curve showed an $IC_{50}$ of 710 nM) and no effect on the control strain. For (**c**), (**d**), (**f**), and (**g**), data presented as mean +/− standard deviation. $n = 3$ biologically independent samples. Source data are provided in the Source Data file.

addition, making it a more convenient and economical solution. Following automated sample preparation and incubation for 16 h, visual inspection of the plates revealed several wells with distinguishable color build-up (Supplementary Fig. 18). Among them, two wells clearly stood out, indicating the presence of $CB_2$ agonists (named AGO1 and AGO2 here, Supplementary Table 3).

We repeated the library screening with the luminescence biosensor strain (KM206) to compare the two methods. The luminescence-based screen required considerable shorter incubation, but also required the addition of the cell lysis reagent and luciferin and detection of luminescence in a plate reader. The results confirmed the hits obtained using the colorimetric strain. As shown in Fig. 5b, AGO1 and AGO2 triggered very strong activation (10- and 80-fold signal over background, respectively). We used the luminescence biosensor to obtain dose–response curves of these compounds and determined $EC_{50}$ values of $389 \pm 61$ nM and $1.5 \pm 0.22$ μM, respectively (Fig. 5c, d).

Screening for receptor antagonists entails competitive inhibition assays where the presence of an antagonist results in the reduction of the response of the biosensor when an activator is also present. Here, accurate quantification of biosensor activity is important. Thus, we chose to use the luciferase biosensor strain. Several potential antagonists were identified by looking for compounds that together with CP55940 resulted in a lower biosensor signal compared to the agonist CP55940 alone (Fig. 5e). From these, we selected seven potential antagonists (ANT1–ANT7) for further experiments and confirmed that these compounds did not exert any non-specific effect on the biosensor, as in all cases the null control strain was fully activated by adenosine. Dose response curves revealed that the most potent antagonists were ANT2 and ANT5 (Supplementary Table 2), exhibiting apparent $IC_{50}$ values of $28 \pm 2.9$ μM and $710 \pm 73$ nM, respectively (Fig. 5f, g). All identified agonists and antagonists (AGO1, AGO2, ANT2, and ANT5) are compounds that have not

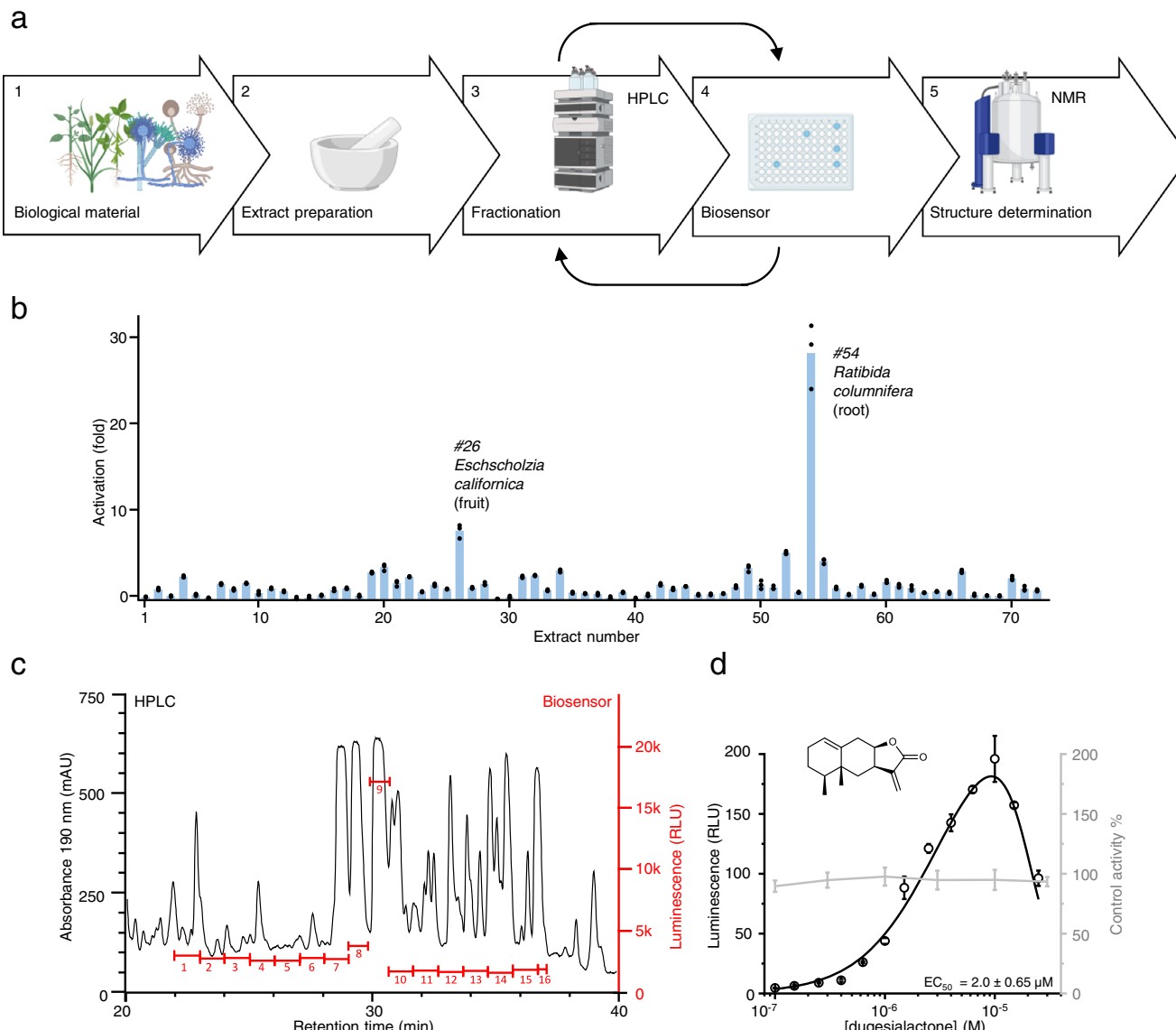

**Fig. 6 Bioactivity-based discovery of GPCR ligands from complex biological samples.** The high sensitivity of the biosensor enables the discovery of GPCR ligands from complex mixtures. **a** The bioprospecting workflow consists of (1) selection of biological material, (2) preparation of extract, (3) fractionation of extract using the chromatographic method of interest, (4) assaying the fractions using the biosensor strain, and (5) structure determination of compounds in the fraction. Depending on the degree of separation of the compounds in the initial fractionation, one or more additional fractionation and biosensor assay steps can be performed to ensure purity of the target compound. **b** Assaying 71 plant extracts with the biosensor strain KM206 revealed two plant extracts that clearly activate $CB_2$. **c** In order to purify main cannabinoid compound from the *R. columnifera* extract, this was fractionated into 16 fractions (red lines) by preparative HPLC (UV trace in black) and each fraction was assayed with the biosensor to find the cannabinoid-containing fractions. **d** The biosensor strain KM206 was incubated with a 10 nM to 100 μM dilution series of dugesialactone (structure determined by NMR in insert). The resulting dose–response curve (black) indicates an EC50 of 2 μM. A control curve produced by incubating dugesialactone with KM207 (gray) shows no non-$CB_2$-specific effect of dugesialactone on KM206 in the specified range). For (**b**) and (**d**), data presented as mean $+/-$ standard deviation. $n = 3$ biologically independent samples. Source data are provided in the Source Data file.

been previously identified to modulate $CB_2$. Furthermore, in order to evaluate the general usability of this high-throughput screening workflow, we calculated Z' scores (measure of statistical effect size)[49] for both the agonist and antagonist screen (Supplementary Table 7). These scores were 0.87 and 0.61, respectively, indicating an excellent assay.

**Case study 2: activity-guided bioprospecting.** To further showcase the capabilities of the developed biosensor, we tested it in a problem with a higher degree of difficulty, screening for active compounds in complex biological samples (Fig. 6a).

Compared to pure compounds (as in the case of compound libraries), the use of biological material usually poses several additional challenges, such as low levels of analytes, and the presence of compounds interfering with different components of the biosensor and the analysis. For this challenge, we chose the biosensor using luminescence as reporter, because colored and fluorescent molecules typically present in plant extracts may mask fluorescent or colorimetric reporter output.

We started by screening 71 methanol extracts from different parts of 54 randomly chosen Mexican traditional medicine plants (Supplementary Table 3). The analysis revealed two plant extracts, namely those from the root of *Ratibida columnifera*

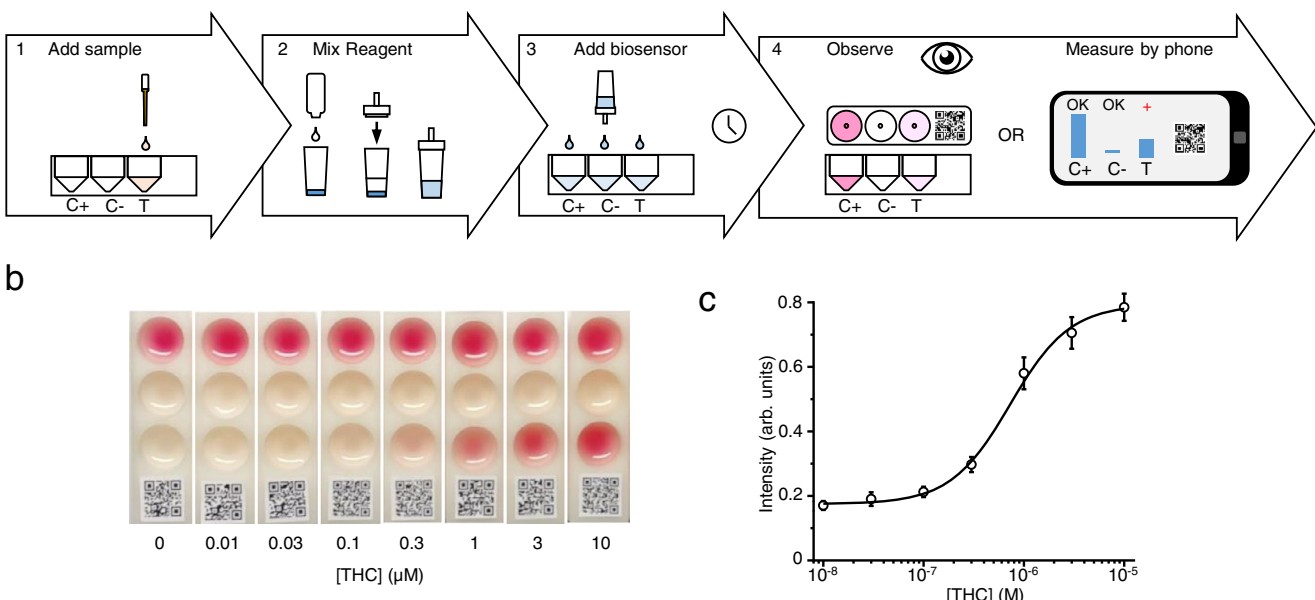

**Fig. 7 The portable colorimetric biosensor. a** The cannabinoid biosensor was configured into a portable biosensor the output of which can be monitored by the eye. This biosensor workflow using the colorimetric strain KM205 consists of (1) Preparing the biosensor strain by activating it with concentrated media (this solution also includes the DOPA and O-da supplement), (2) adding the sample to be measured in the portable biosensor device, (3) adding the activated biosensor yeast to the device and incubating 4–16 h, (4) observing the color that was produced or measuring the signal by cell phone camera. **b** To test the portable biosensor workflow, a concertation series of CP55940 from 30 pM to 300 nM was added to the test wells (T) in of ten biosensor devices. Positive control wells (C + ) in each device had 300 nM CP55940 each and no negative control wells (C−) no cannabinoid. After the addition of the biosensor and 16 h of incubation, the color in test wells could be detected by eye down to 30 pM CP55940. **c** To quantify the biosensor signal for the portable colorimetric biosensor the above experiment was made in triplicate and the O-da-betacyanin pigment measured from a cellular phone photograph (see "Methods"). This resulted in a typical dose–response curve. For (**c**), data presented as mean + / − standard deviation. $n = 3$ biologically independent samples. Source data are provided in the Source Data file.

and the fruit of *Eschscholzia californica*, standing clearly above the background, showing a 27-fold and 7-fold increase in signal respectively (Fig. 6b). We prioritized *R. columnifera* and used again the biosensor to perform bioactivity-guided fractionation in order to identify the active cannabinoid compounds from the extract. First, the extract was divided into 16 fractions by preparative High-Performance Liquid Chromatography (HPLC) and dilutions of each fraction were assayed with the luciferase biosensor strain. The highest activity was present in fraction 9 (Fig. 6c and Supplementary Table 5). Therefore, this fraction was further separated by preparative HPLC (Supplementary Fig. 19) and, finally, the purified compound was submitted to (nuclear magnetic resonance) NMR analysis (Supplementary Table 6). This revealed that the active compound is the sesquiterpene lactone, dugesialactone (Fig. 6d). Dugesialactone was previously isolated from another Mexican plant, *Dugesia mexicana*[50,51], but its function as phytocannabinoid was never identified. The dose–response curve of dugesialactone was found to have a bell-like shape and an apparent $EC_{50}$ of 2 µM for the first phase (Fig. 6e). To rule out that this behavior results from interference of dugesialactone with non-receptor components of the biosensor at high concentrations, we tested dugesialactone with the null control strain (KM207) and found no interference (Fig. 6e).

**Case study 3: point-of-use portable biosensors.** In order to expand this technology into consumer and non-specialist applications, a portable biosensor that does not require the use of lab equipment or expertise is needed. In this case, the biosensor must be configured so that it can be read by the eye or with common

devices such as a cell phone. Thus, we developed two portable biosensor devices, one for color measurements using the betalain reporter strain, and another for light measurement using the luminescence reporter strain, and developed workflows for the detection of cannabinoids in real-life samples.

First, we developed a workflow for the colorimetric strain KM205 that uses a dedicated bar-coded device for easy sample handling (Fig. 7a). According to this workflow, first, the sample is applied to the sample well, and then the biosensor is mixed with the activating reagent (concentrated minimal media) and applied to the dedicated device (test sample; S) (Fig. 7a). The control wells contain either a known amount of a dried cannabinoid (positive control; C + ) or no cannabinoid (negative control; C−). After incubation for 4–16 h, the activated biosensor can be readily read by eye (Fig. 7b).

To further improve analysis with this system, we also developed a method for quantifying cannabinoids using a cell phone camera and RGB color analysis. The biosensor output is measured from a photograph as the average redness (R) of the sample (ratio of red channel to green and blue channels, see methods). A dose–response curve for CP55940 measured by cell phone (Fig. 7c) is consistent with one measured by a plate reader (Fig. 3d). The Z' score[49] for cannabinoid detection using a cell phone was calculated to be 0.716, qualifying it as an excellent assay. The LOD for THC was 100 nM, indicating similar sensitivity with other assays described in this work.

Whereas the colorimetric biosensor strain is well suited for the qualitative detection of cannabinoids from samples, some portable applications, such as regular monitoring of cannabinoids in samples from patients treated with medicinal cannabis, require

the sensitive, semi-quantitative measurement of compounds and a shorter development time. Thus, we developed a portable biosensor based on the luminescence reporter strain and a device that utilizes a cellular phone and can be assembled from readily constructed or purchasable parts (Fig. 8a). This workflow consists of first adding the sample to the sample well and then mixing the biosensor yeast with the activating reagent and applying it to the device. Following incubation, the developing solution is added in all wells. The biosensor device is assembled by attaching an adaptor ring and a clip-on cell phone macrolens (here, Xenvo Clarus 15x) creating a closed, standardized, environment for image acquisition (Fig. 8b). Finally, the sample image is acquired and analyzed by software that quantifies biosensor output by measuring intensity of the blue channel of the resulting raw picture. By measuring the signals from the five calibration wells, a reference curve can be created by the software and used to estimate the concentration of the target compound in the sample.

To evaluate the performance of the portable biosensor, we generated dose–response curves for THC (Fig. 8c), its liver metabolism product 11-OH-THC (Fig. 8d), and the synthetic cannabinoid JWH-018 (Fig. 8e). The portable biosensor-derived curves were consistent with those measured in a laboratory setting with a luminometer (plate reader). LODs for THC, 11-OH-THC, and JWH-018 were 100 nM, 100 nM, and 10 nM, indicating similar sensitivity as other assays from this work. Moreover, Z' values for the determination of THC, 11-OH-THC, and JWH-018 were found to be 0.89, 0.83, and 0.825, respectively, qualifying the assay as excellent[49]. Next, to evaluate the (semi) quantitative use of the portable biosensor, the concentration of a cannabinoid-containing sample was determined by analyzing it with the biosensor and comparing the resulting signal with that of calibration standards. For all three compounds tested, the biosensor was able to correctly estimate the concentration in the sample; i.e., the sample containing 150 nM THC produced a reading placing it between 100 and 300 nM THC calibration points, the sample with 150 nM 11-OH-THC was estimated to contain between 100 and 300 nM of the cannabinoid, and the sample with 15 nM JWH-018 was correctly placed between the 10 nM and 30 nM standards (Fig. 8f). Finally, we demonstrated the usefulness of the portable biosensor for real-life applications using simulated human samples. Following cannabis consumption, THC and/or its metabolic products 11-OH-THC and 11-oxo-THC can be found in urine, saliva and blood in nano- to micro-molar concentrations[52]. To simulate cannabinoid-containing human samples, artificial saliva, urine, and calf serum were supplemented with THC according to concentrations typically found in authentic samples[53–55] and assayed using the portable biosensor. Comparing the signal from these samples to corresponding controls without cannabinoids showed clear detection of THC (Fig. 8g).

Overall, we developed two efficient, economic, and robust portable biosensor assays based on custom-made, easy-to-use devices, which involve simple workflows that can be carried out by a non-expert using a standard cell phone. The characteristics of the colorimetric assay make it well suited for facile parallel processing of large numbers of samples required for mass testing, e.g., quality control of cannabis products or screening samples at customs or sports events. The luminometric assay, on the other hand, allows for estimating in a short time the concentrations of cannabinoids in urine, saliva, and serum samples.

## Discussion

In this work, we demonstrate proof-of-concept for the use of GPCRs as the sensing units in yeast-based whole-cell biosensors for specific biotechnological applications. Earlier work established a general framework for integrating heterologous GPCR signaling in yeast[5,31,28,29]. In order to make GPCR-based biosensors a cost-efficient, sensitive, and widely applicable method, here, we introduce technical improvements and associated portable devices and workflows that enable challenging applications.

The first application we demonstrate is chemical compound library screening. GPCRs are very important as drug targets, and 34% of FDA-approved drugs act on GPCRs[56,57]. $CB_2$, in particular, is the target of multiple ongoing drug discovery efforts[58]. Our $CB_2$ biosensor displays sensitivity and robustness on par with that of mammalian systems[24], which are clearly more costly, complicated to use, and require specialized equipment, infrastructure, and trained personnel. Our technology has the potential to democratize GPCR drug screening, making it available to small research labs or limited-budget companies. The developed biosensor strains are available to the research community and will enable non-specialized laboratories, such as those working in chemical biology of natural products discovery, to identify $CB_2$ ligands with a simple workflow. The modular design of the basic strains constructed here will facilitate further development of more or improved biosensors by the research community, for example, for different GPCRs of interest. This is particularly relevant today, as several publicly available repositories for new compounds have been established in the past years[59]. Our biosensor will not replace assays with mammalian cells, because these are indispensable for functional validation, but the developed yeast-based biosensor can help narrow down large libraries into a subset of hits that can be further evaluated using mammalian systems. Furthermore, although mammalian cell systems can be affected by off-target effects of drug leads, a yeast-based system can help focus on the effect of a drug directly on the receptor, thus providing complementary information. The low background of the yeast system and its high dynamic range can also be beneficial for detecting more subtle effects of potential pharmacophores that can be subsequently further optimized by chemical functionalization.

Subsequently, we address a more complex challenge, by demonstrating the use of the cannabinoid biosensor in bioprospecting. For this application, developing biosensor with high sensitivity and low background is paramount, as the compounds of interest are frequently present in low concentrations and mixed with compounds that can potentially interfere with both the receptor and the reporter. Another challenge is the potential toxicity of target or non-target compounds against the cell line. As a yeast-based system, our biosensor is less likely to experience toxic effects than animal cells. Using this biosensor, we identified a phytocannabinoid, dugelsialactone, a compound that was previously known as an anticancer lead[60], but had never been demonstrated to be a cannabinoid.

Finally, we demonstrate the development of portable biosensor devices that can easily detect the presence of cannabinoids in real-life samples outside the lab and without the need for expensive equipment or trained personnel. This requires biosensors that are sensitive, robust, and with an output that can be measured by eye or common equipment such as a cell phone camera. Contrary to available quick tests that are based on antibodies and can only detect a specific cannabinoid, our sensor is able to detect any of the structurally diverse cannabinoids that is a $CB_2$ ligand. Examples of such applications include monitoring endocannabinoid disease biomarkers[61] or residual cannabinoids after therapeutic treatment with cannabinoids from human urine, saliva or serum samples, quality/potency control and breeding of medical cannabis, or interception of dangerous designer drugs (synthetic cannabinoids). We envision that the portable biosensors can be further improved by optimizing the cell phone accessory devices and software. Currently, the luminescence-based biosensor can

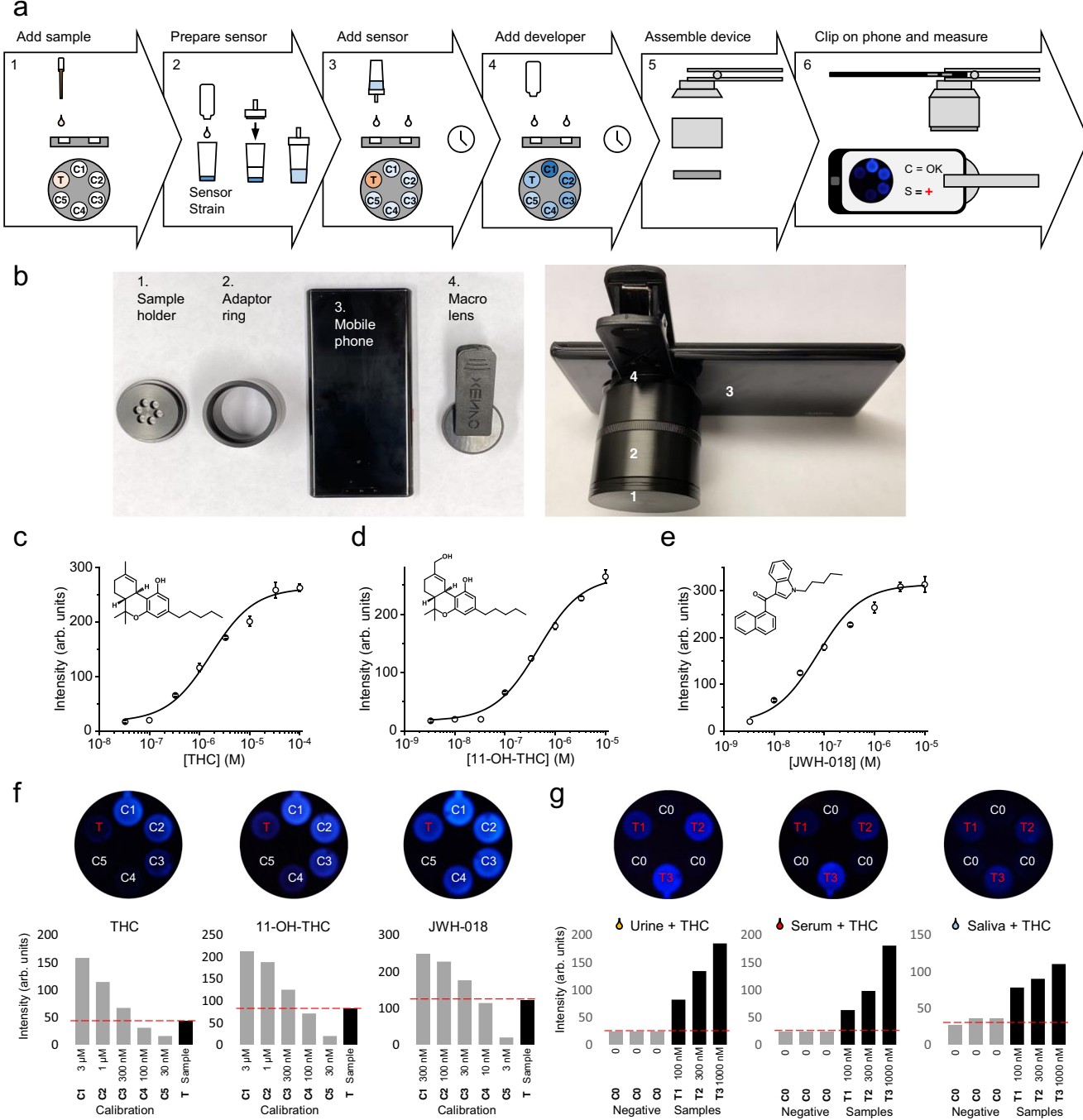

**Fig. 8 The portable luminometric biosensor. a** The cannabinoid biosensor was configured into a portable biosensor that can be read by common equipment such as a cell phone. This biosensor workflow using the strain KM206 consists of: (1) preparing the biosensor strain by activating it with concentrated media, (2) adding the sample to be measured in the portable biosensor device, (3) adding the activated biosensor yeast to the device and incubating, (4) adding the developer solution (lysis buffer and luciferin), (5) assembling the rest of the portable biosensor device with a cellular phone, and (6) measuring biosensor output with the phone. **b** The portable biosensor device is assembled by attaching the sample holder disk (1) containing the activated biosensor to the adaptor ring (2) and the macrolens (4). This can then be fitted on the standard cell phone camera (3). **c** To validate the workflow with the device, dose–response curves were produced by incubating the strain for 3 h in the device in the presence of 30 pM–100 μM of the canonical cannabinoid THC. **d** Dose response obtained with 3 pM–10 μM of 11-OH-THC. **e** Dose response obtained with 3 pM–10 μM of the synthetic narcotic JWH-018. **f** Evaluation of cannabinoid concentrations by measuring light intensity in the sample well (T) and comparing with calibration points (C1–C5) of the respective compounds. Here, 150 nM THC, 150 nM 11-OH-THC or 15 nM JWH-018 (black bars) show signal intensities placing them between calibration points for 100 nM and 330 nM THC, 100 nM and 330 nM 11-OH-THC, and 10-33 nM JWH-018 (gray bars), respectively. **g** Detection of cannabinoids in real-life-like samples. KM206 was used to analyze reconstituted urine, serum, and saliva samples containing 100 nM (T1), 330 nM (T2), and 1 μM (T3) THC (black bars). KM206 without THC was used as negative control (C0) (gray bars). For (**c–e**), data presented as mean +/− standard deviation. n = 3 biologically independent samples. Source data are provided in the Source Data file.

generate a measurable signal already after 30 min. Exploiting this speed fully will require utilization of the full sensitivity of cell phone cameras, improved optics, and additional signal processing. In principle, the portable biosensor can detect any $CB_2$-modulating compound. However, we believe that biosensors that can distinguish between similar cannabinoids could be developed by engineering $CB_2$ (or $CB_1$) using a directed evolution approach[62] or by sourcing homolog GPCRs with desired properties from other organisms. Finally, we foresee that the portable biosensor technology and platform strain can be used to enable detection of numerous other biomarkers using other GPCRs, thus paving the way for smart living diagnostics.

In addition to the three case studies demonstrated here, our GPCR-based biosensor platform can be expanded to fit numerous other applications. For example, there is a strong potential for using the biosensor in metabolic engineering efforts to monitor and optimize strains for the bioproduction of small molecules or enzymes. Moreover, GPCR-based biosensors can be used to screen mutant libraries for enzymes with improved activity or to drive directed evolution of enzymes.

# Methods

**Chemicals and enzymes**. All chemicals were reagent grade. CP55940, L-DOPA, O-diadizidine, 5-FOA and the CeLytic Y reagent were purchased from Merck Life Science A/S (Søborg, Denmark). The compounds corresponding to HTS screening hits AGO1 (#6118258), AGO2 (#5143116), ANT1 (#5169183), ANT2 (#5182514), ANT3 (#5177027), ANT6 (#5377528), ANT7 (#5148134) were obtained from ChemBridge, US. ANT4 (#3888-1206) and ANT5 (#3909-7498) were bought from ChemDiv, US. 11-OH-THC and JWH-018 were obtained from LGC Standards Ltd., UK And THC was bought from Chiron AS., Norway. Nano-glo luciferase reagent was bought form Promega, US. The chemical library used for high-throughput screening was obtained from the Chemical biology and HTS facility of the University of Copenhagen (https://cbhts.ku.dk/chemical-compound-collection/). Restriction enzymes, and digestion and PCR reaction buffers were from New England Biolabs (NEB), USA. Pfu-X7 DNA Polymerase[63] was made in-house. MangoMixTM 2x Taq DNA polymerase PCR master mix was purchased from Meridian Life Science Inc., USA.

**Plant material**. Plant material for the 49 plant species was collected from the Botanical gardens, grown from seeds or collected in the wild; details are given in Supplementary Table 4. Botanical identification was ensured by MSc. Eduardo Blanco Contreras curator at the Centro de Referencia Botanica de la Universidad Autonoma Agraria Antonio Narro (CREB-UAAAN) where vouchers of the specimens were deposited.

**Yeast strain construction**. Yeast strains used in this study are listed in Supplementary Table 1. All strains were produced by genomic integration of DNA fragments (linearized plasmids or PCR products) transformed into the relevant strain either by the lithium acetate method[64] or by electroporation. Chassis strains were produced by HR-based knockout[65] and the subsequent strains by HR-based modular multi-part integration[37] (Supplementary Fig. 3).

**PEG-lithium acetate mediated transformation**. Yeast transformations were performed according to a modified PEG-LiOAc protocol[64]. A 5 mL saturated overnight YPD culture was diluted to 0.25 OD and grown until 1 OD. Then the yeast was pelleted by centrifugation and washed with 0.1 M LiOAc. The cells were then resuspended in 20 μL of 0.1 M LiOAc followed by the addition of 10 μL of heat-denatured salmon sperm DNA and 200 μL of PLI solution (45% polyethyleneglycol-3350 with 0.1 M LiOAc). For each transformation, 200 μL of this cell stock was mixed with linear plasmid DNA to be transformed and heat-shocked at 42 °C for 30 min. Then, the cells were washed with $H_2O$ and plated on CM-U plates.

**Electroporation**. Yeast cells were inoculated in YPD and grown overnight at 30 C with 150 rpm shaking to a saturated culture. Then, the cells were diluted into 50 mL of YPD to $OD_{600} = 0.25$ and grown until $OD_{600} = 1$. The cells were collected by centrifugation for 5 min at 4 °C and 3000×$g$ and resuspended in 10 mL ice-cold $H_2O$. Next, the cells were treated with EP solution (1 M sorbitol, 100 mM LiOAc. 10 mM HEPES (pH 7), 10 mM DTT) for 15 min at 30 °C. The cells were washed twice with ice-cold 1 M sorbitol and resuspended in 0.2 mL of ice-cold 1 M sorbitol. In total, 5 μL PCR product was added to 40 μL of yeast cell suspension in a 0.2-cm electroporation cuvette. The cells were electroporated with a BIO-RAD Gene Pulser Xcell using the standard protocol for *S. cerevisiae* (25 μF, 200 ohm, 1.5 kV). The cells were resuspended in 1 mL 1 M sorbitol and incubated for 1 h at 30 °C. Finally,

the cells were collected by centrifugation for 1 min at 8000×$g$ at RT, resuspended in 100 μL of 1 M sorbitol, and plated on selection plates (CM-U).

**Generation of knockout mutant strains**. Chassis strains were produced by stepwise knocking out *STE3*, *SST2*, *FAR1*, *STE12*, and *GPA1*. Briefly, the parent strain was transformed with either knockout cassette (Supplementary Table 9) containing the URA3 selectable marker flanked by LoxP sites. This fragment was PCR amplified from the plasmid pUG72[65] using primers containing 40–50 bp overhangs corresponding to the genomic sequence of the KO target gene. The transformed yeast was then plated on yeast selection media (CM-U). Knockout cassette-positive colonies were identified by PCR yeast genotyping using the primer 44 (B-M-R) combined with corresponding primers 46–55 (Supplementary Table 10). Subsequently, a knockout-positive yeast clone was transformed with a plasmid conferring galactose inducible CRE-LOX recombinase expression. Finally, the URA3 selection marker was removed by CRE-LOX recombination and a markerless clone was picked via 5-FOA counter-selection. The procedure was then repeated with the next knockout. For the GPA1 knockout, electroporation was used to ensure a high enough transformation rate of the KO cassette. After the last knockout, the CRE-LOX plasmid was removed by curing.

**Plasmid construction**. Plasmid constructs used in this study are listed in Supplementary Table 8. All plasmids newly constructed for this study were made with USER cloning[66].

**USER cloning**. Promoters, ORFs and other parts (Supplementary Table 9) were PCR amplified with USER primers (Supplementary Table 10). Plasmid vectors were prepared for using cloning by digesting them with AsiSI and Nb.BsmI restriction enzymes and USER-PCR-fragments were ligated into USER vectors according to ref. [38].

**Primers**. All PCR Primers used in this study are listed in Supplementary Table 10. Primers were ordered from TAG Copenhagen, Denmark.

**Yeast membrane preparation and western blot analysis**. (Enriched) plasma membrane preparations of strains were produced by differential centrifugation[67]. Briefly, the cells from an overnight yeast culture in YPD were collected by centrifugation at 3000×$g$ washed with water and ice-cold lysis buffer (100 mM Tris-HCl pH 7.5, 10 mM EDTA pH 8, 1 M sorbitol, 1 mM PMSF, 1 μM pepstatin), resuspended in ice-cold lysis buffer and lysed by vortexing for 10 min at +4 °C with glass beads. After removing the glass beads the sample was centrifuged at 1000×$g$ at 4 °C for 10 min to remove unbroken cells and coarse cell debris. Then the supernatant was centrifuged at 20,000×$g$ at 4 °C for 20 min. Subsequently, the pellet was resuspended in solubilisation buffer (100 mM Tris-HCl pH 7.5, 60 mM NaCl, 20% glycerol, 5 mM $MgCl_2$) by pipetting and using a potter. Protein concentration of samples were determined using Bradford method.

For western blot analysis membrane samples containing 40 μg of protein were prepared in protein loading buffer containing sucrose and separated on a Bio-Rad stain-free gel in TGS buffer. Total protein was visualized using a Bio-Rad gel imager, then proteins were blotted on a membrane PVDF membrane using the Bio-Rad Trans-Blot Turbo system. The membrane was blocked with 0.5% skimmed milk in TBST for 1 h and then primary antibody (anti HA-Tag (C29F4) Rabbit mAb, Cell Signaling) was added 1:1000. After 1 h the membrane was washed with TBST and the secondary antibody (Polyclonal Swine Anti-Rabbit Immunoglobulins/HRP, P021702-2 DAKO) was added 1:1000. The membrane was washed with TBST and luminol solution was (ECL Western blotting substrate, Pierce) added and luminescence visualized with a Bio-RAD ChemiDoc MP imaging system using the Image Lab 5.2.1.

**Fluorescence microscopy**. Cells were observed under Olympus BX60 fluorescence microscope using GFP settings (ExF: 465–500 nm, DM: 505 nm, BF: 516–556 nm) with AnalySIS docu 5.0.

**Induction of biosensor strains**. Biosensor strains were grown until saturation. Then cells were pelleted by centrifugation and cell density set to $OD_{600} = 5$ (luciferase strains) or 0.5 (betalain or ZsGREEN strains) by resuspending in fresh CM-U media. In total, 20–200 μL of cells were dispensed in 96-well plates and an inducer was added. The cells were then induced for 15 min to 24 h at 30 °C with 200 rpm shaking. For betalain reporter strains the additives 0.1 mg/mL tyrosine, 0.1 mg/mL L-DOPA, or 0.5 mM o-Diadisidine were added at this stage if relevant.

**Analysis of THC content in biosensor cells**. Overall, 10 μl 1 mM THC in DMSO or DMSO was added to the strain KM206 at density $OD_{600} = 5$ in fresh CM-U media, and cells incubated 3 h at 30 °C. Then, cells were collected by centrifuging at 3000×$g$ for 5 min. An extraction control was produced by adding the equivalent amount of THC to a cell pellet of non-treated treated cells. Subsequently, the cells were lysed with glass beads in methanol and cleared by centrifugation at 20,000×$g$ for 15 min and filtered through a 0.22-μm filter. Finally, the THC content was

measured using HRMS. The HRMS data were acquired on a Dionex UltiMate® 3000 Quaternary Rapid Separation UPLC-focused system (Thermo Fisher Scientific, Germering, Germany) coupled with a Bruker Daltonics Compact QqTOF mass spectrometer equipped with electrospray ionization (ESI) interface (Bruker Daltonics, Bremen, Germany). The separation was performed at 40 °C (flow rate 0.3 mL/min) using a Phenomenex Kinetex XB-C18 column (100 mm × 2.1 mm i.d., 1.7-μm particle size, 100-Å pore size) (Phenomenex, Inc., Torrance, CA, USA) with the following gradient profile: 0 min, 2% B; 15 min, 100% B; 18 min, 100% B; 19 min, 2% B; 26 min, 2% B. The mobile phases were water (A) and 100% acetonitrile (B), both containing 0.05% formic acid. The MS data were acquired with the following setting: capillary voltage, 4500 V; end plate offset, −500 V, drying gas flow, 8 L/min; drying temperature, 250 °C; nebulizer pressure, 1.2 bars.

**Fluorescent reporter measurements**. The ligand response of the cannabinoid biosensor strains with ZsGreen reporter gene were measured using a Molecular devices SpectraMax-M5 plate reader using SoftMax Pro 6.2.2 software. In all, 100 μL of induced cells were added to black clear-bottom 96-well microplates and the fluorescent signals were read using a 480 nm excitation, 495 nm cutoff, and 515 nm emission wavelength.

**Betalain reporter measurements**. The signal from the betalain reporter was measured using a Molecular Devices SpectraMax-M5 plate reader and SoftMax Pro 6.2.2 software with 470 nm and 520 nm absorption wavelengths for betaxanthins and betacyanins, respectively.

**Luciferase reporter assays and measurements**. Luciferase signal measurements were made by mixing 25 μL of induced cells together with 25 μL luciferase reagent (CeLytic Y with 4% Nano-glo reagent) in a black ProxiPlate™ Perkin-Elmer (#6006270). After 10 min of incubation, luminescence was measured with Molecular Devices SpectraMax-M5 plate reader and SoftMax Pro 6.2.2 software with 0.5 s integration time.

**Data analysis**. Luminescence, fluorescence or colorimetric data was analyzed in OriginPro 2020 9.7.0.188 software. Curve fitting was performed using the sigmoidal fitting Hill1 or biHill algorithm with default settings. LOD was determined as a lowest experimental measurement that was significantly different ($t$ test, two-sided, $P < 0.05$) from the negative control (no ligand).

**High-throughput screening**. For each HTS experiment 100 μL or 40 μL of cells (see the induction of biosensor strains) were dispensed into 96-well or 384-well plates using an OT-2 pipetting robot. Then, 1 μL of the chemical library was added to these cells. After incubation (16 h for betanin strain, 3 h luciferase strain) biosensor output was evaluated by eye (betanin strain) or by mixing in 20 μL of developer solution (Celytic Y reagent with 1% Nano-glo reagent) and luminescence measurement with SpectraMax-M5 plate reader using SoftMax Pro 6.2.2 software.

**Extraction of plant material**. Dried plant material was ground to a fine powder using a domestic grinder. Subsequently, powder was suspended in Methanol at a ratio of 1:10 (w/v) and ultrasonicated for 30 min at room temperature (24 °C) and filtered under reduced pressure. The resulting extracts were then dried by rotor evaporation, the dry weights were recorded, and dried crude extracts were re-suspended in DMSO to a concentration of 50 mg/mL.

**Isolation of dugesialactone**. The isolation of dugesialactone was performed with a Shimadzu Prominence LC-20A system, consisting of a SIL-10AP autosampler, a LC-20AT quaternary pump, a CTO-10ASvp thermostatted column compartment, a SPD-M20A diode array detector, and a FRC-10A fraction collector using Lab-Solutions 5.71 software. Around 20 consecutive injections of crude EtOAc extract (0.2 mL per injection, 50 mg/mL in MeOH) were separated at a flow rate of 2 mL/min using the above-mentioned solvents with a Phenomenex Luna C18 (2) column (250 × 21.2 mm, 5μm, 100 Å; Phenomenex, Torrance, CA, USA). Separations were performed using the following elution profile: 0 min, 10% B; 30 min, 100% B; 50 min, 100% B.

**NMR experiments**. NMR data were acquired in automation with temperature equilibration to 300 K, optimization of lock parameters, gradient shimming, and setting of receiver gain on a Bruker Avance III 600 MHz NMR spectrometer equipped with a Bruker SampleJet sample changer and a cryogenically cooled 1.7-mm TCI probe head. NMR sample change and data acquisition were automatically controlled by IconNMR ver. 4.2 (Bruker Biospin, Karlsruhe, Germany). Topspin ver. 3.5 (Bruker Biospin, Karlsruhe, Germany) was used for processing NMR data.

**Portable biosensor experiments**. Portable biosensor devices (Figs. 7 and 8) were constructed by Flemming Frederiksen and team at the Workshops, Department of Plant and Environmental Sciences, University of Copenhagen. Betalain portable biosensor experiments were performed by dispensing 0.5 mL of KM205 biosensor strain culture (see the induction of biosensor strains) in each well (Fig. 7) and

adding 5 μL of 100× inducer. The devices were covered with a plastic film and incubated for 16 h at 30 °C with 200 rpm shaking. After incubation, the devices were visualized by eye and photographed. Betalain reporter output was quantified using a cell phone camera by analyzing it in ImageJ 1.53 C (with FIJI plugin pack) and calculating the redness of each sample as previously described in ref. [28]. Briefly, redness was determined as (the red channel signal minus the average of the blue and green channels) divided by the red channel signal.

Luciferase portable biosensor experiments were performed by dispensing 37.5 or 50 μL of KM205 biosensor strain culture (see the induction of biosensor strains) in each well (Fig. 8) together with 1 μl of 100× inducer or 12.5 μL of artificial saliva with mucin (Pickering #1700-03169) spiked with either cannabinoid and incubating for 3 h at RT. After incubation 50 μL of developer reagent (Celytic Y reagen with 4% Nano-glo reagent) was added to each well and incubated 10 min at RT. Then, the device was assembled by adding the adaptor part (Fig. 8), macrolens (Xenvo Clarus 15×) and attaching the assembly a Huawei Mate 30 pro cell phone (Android 10). RAW format photographs were taken in pro mode using 1 s or 30 s of exposure time, 1000 or 6400 ISO setting, EV = 0, auto focus-F, and auto white balance settings. Since the peak wavelength of NanoLuc (450 nm) is the same as that of a typical bayer filter, Intensity of luminescence was quantified from the RAW picture by measuring intensity of the blue channel only. The RAW pictures were analyzed with the RawTherapee 5.8 open-source software. To isolate the blue channel no demosaicing was chosen, and the red and green channels were disabled. The blue intensity was measured with the analysis tool.

**Reporting summary**. Further information on research design is available in the Nature Research Reporting Summary linked to this article.

## Data availability
Data supporting the findings of this work are available within the paper and its Supplementary Information files. Additional files containing raw images generated in this study have been deposited at Zenodo.org (https://doi.org/10.5281/zenodo.6587270). Source data are provided with this paper.

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

## Acknowledgements

We would like to thank Dr. Renaud Wagner (Institut de Recherche de l' Ecole de Biotechnologie de Strasbourg) for providing the CB$_2$ clone, Flemming Frederiksen and his team (University of Copenhagen) for the construction of portable biosensor devices, and Marek Miettinen for assistance with portable biosensor imaging and analysis development. We would also like to thank Dr. Simon Dusséaux (University of Copenhagen, Denmark) for critically reading the manuscript. The project leading to this application has received funding from the European Union's Horizon 2020 research and innovation programme under the Marie Sklodowska-Curie grant agreement No. 845835. This work was also supported by the Novo Nordisk Foundation grants NNF16OC0021760 and NNF19OC0055204 (to S.C.K.), NNF17OC0027646 (to S.B.), NNF18OC0031872 (to K.M.), and NNF16OC0021616 (to D.S.), and the Danish Council for Independent Research grants 7017-00275B (to S.B. and S.K.) and 0136-00410B (to S.C.K.). Elements featured in some figures of this manuscript were created with BioRender.com.

## Author contributions

K.M.: conceived the project, designed research, constructed yeast strains, performed biosensor experiments, analyzed the data, and wrote the manuscript. N.L.: constructed yeast strains, performed biosensor experiments, and analyzed the data. L.R.H.: constructed yeast strains, performed biosensor experiments, and analyzed the data. A.R.: conceived the bioprospecting application, sourced and prepared plant material, performed biosensor experiments, performed chromatography, and analyzed the data. Y.Z.: performed compound purification, NMR data analysis, and structure elucidation. I.E.N.: performed biosensor experiments. J.B.A.: assisted with HTS analysis. M.G.: assisted with HTS analysis. D.S.: NMR data analysis and structure elucidation. S.B.: designed research. S.C.K.: conceived the project, designed and coordinated research, and wrote the manuscript.

## Competing interests

The authors declare no competing interests.
