## [Peer Review File · Nature Communications]

A GPCR-based yeast biosensor for biomedical, biotechnological, and point-of-use cannabinoid determinationReviewers' Comments:

Reviewer #1:

Remarks to the Author:

The innovative idea is to clone human or humanized GPCR CB2 into yeast to mimick G coupled receptor signal transduction into yeast. The first question here, why did the authors choose CB2 and not CB1? The system is used as a biosensor system to detect CB2 mimicking compounds or drugs. The system is build up in a modular way where Gbeta and Ggamma have nor been protein engineered. The detection unit is coupled to various actuator modules and all have been discussed in detail.

The work is significant, beacause it gives a new insight in a bioassay detector system.This may provide a new approach for High Throughput Screening.

Advantage of the system is the flexibility by design to adapt various GCPR like structures for various potetnial ligands. THis is an important feature because of allosteric modulation, but that has not been adressed in this manuscript. Here, the authors must discuss limitations of the system due to many allosteric sites in GPCR have an impact on signal transduction.

It is not clear where the GPCR CB2 is located? What organelle, what spatial resolution in the yeast cell? Here, more data like Ab immunofluorescence detection is needed.

Cannabinoids show very low soulubility and accumulate by more than 50% into membranes. Does this have an impact on the bioassay? Are the detemined EC50 correct, if most of the drug does not enter intracellular GCPR CB2 receptors?

Why do we see totally different kinetic data for induced luminescence in Fig. 4? If this is an CB2 receptor, why did the authors not choose CBD or another CB2 specific lignd? THis brings up the question, if we can distinguish between a full an partial agonist / antagonist?

I like the idea of a portable CB2 detector and recording data with a camera, but I amnot sure if this makes sense to get valid data out of a crude mixture of a biological sample.

Overall an intersting report with a new concept of using heterologous GCPR CB2 receptorrrs. But, we should be aware that cannabinoid like effects ae orchestrated by CB1, GPR55 and COX2 in parallel. THERefore a yeast based systems will come fast to its limitations.

The conclusions drawn are fine, but there is still much space for improvement. TO give one example, the EC50 rang is in the micromolar range what is for a new modern drug far too high. Here, the nanomolar range must be the future target. THE methodology sounds well and the Danish group is known for solid work in the field of analytics.

Reviewer #2:

Remarks to the Author:

This manuscript describes the development of a yeast-based reporter gene system for measuring the activity of recombinantly expressed GPCRs. While this system could presumably be used for any recombinantly expressed GPCRs, CB1 and CB2 cannabinoid receptors were used for the current work. This work builds on previous work that has defined GPCR pathways in yeast, which the authors cite appropriately. The authors optimize this assay system by testing several reporter gene outputs (Luciferase is found to be best), and use it for screening both a 1600 compound library of small molecules and 71 samples of natural product extracts vs CB2. The assay is shown to be usable for screening, with active compounds identified from the screening. Importantly, evidence was provided that these hits were "on target" using a "null assay" (same assay but without the recombinant CB2 receptor). Finally, the authors use this assay technology to develop a portable detection kit for detecting cannabinoids from body fluid samples. An advantage of this kit over current cannabinoid detection technologies is that it can detect any kind of CB1 agonist based upon its cellular activity, and therefore is not limited to identifying compounds with known

cannabinoid structures.

It was difficult for me to identify precisely where the major scientific advancement is meant to be. Cell-based reporter gene assays are certainly not new. Typically they are not favored for high-throughput screening (HTS), as they are prone to a high rate of false positives because there are many points downstream of the target GPCR where compounds could act to modulate expression of the reporter gene. Therefore, other cell-based assays that measure more proximal events (changes in second messenger levels such as Ca²⁺ or cAMP) are more often employed for GPCR HTS. Reporter gene assays (such as the mammalian PrestoTango assay system) are more typically reserved for "problematic" GPCRs, such as orphan GPCRs where the relevant second messengers are unknown. For researchers who are experienced with reporter gene assays, it will certainly come as no surprise that the luciferase version led to the best assay compared to a fluorescent protein or colorimetric readout. The researchers claim that their assay can "democratize drug discovery by enabling low-cost high-throughput analysis using open-source automation." Yes, open-source automation has significantly reduced the cost of setting up an HTS laboratory; but that is completely independent of this paper. This work does not enable or expand the impact of this cheaper lab automation in any way; this assay benefits from the less expensive equipment the same way that any assay would. Although the authors claim throughout the paper that their assay provides a "far more economical and user-friendly format" for HTS, there is no explanation of how exactly. Likely the authors are referring to the fact that yeast are somewhat less expensive to grow and maintain than mammalian cells, but this represents marginal savings in the overall big picture of an HTS campaign; and the HTS campaign itself (typically \$50-100K, depending upon size of the library and other factors) is a relatively insignificant portion of the \$10-15M it takes to bring a new compound into the first phase of clinical trials. This assay will hardly democratize drug discovery. Furthermore, it has significant disadvantages: 1) a great deal of genetic engineering of the yeast is required, which involves its own money and time cost. 2) It utilizes a yeast-based cell system, rather than mammalian, taking the GPCR target one step further away from its native, physiologically relevant environment. 3) Physiological relevance (at the molecular level) is further reduced by the fact that a) the yeast's natural G protein regulator protein (SST2) is knocked out to increase signaling, b) the G protein used is a chimeric yeast/mammalian mutant G protein, and c) the GPCR itself is artificially fused to another (yeast) protein, with unknown consequences for receptor 3-D structure and function. Whatever the authors' view about the recent trend toward more physiologically relevant screening assays may be (I personally feel that this is in many cases taken too far), we all must recognize that this trend does exist. Even mammalian cell-based assays that utilize a mutant G protein in order to couple GPCR signaling to a more convenient readout (use of Gqi5 mutant G protein to artificially couple GPCRs to Ca²⁺, for instance) have largely fallen out of favor as drug discovery researchers have begun putting greater and greater emphasis on physiological relevance, even for the assays used for HTS. In short, I see a number of potential disadvantages to using this or a similar assay system for HTS, with the only potential advantage being a relatively small cost savings/increase in convenience that could come from using yeast vs mammalian cells. I honestly don't see this assay getting much use as an HTS assay.

Perhaps the real novelty/innovation of this manuscript is meant to lie in the portable cannabinoid detection device that the authors have created. I must admit that while I am an expert in HTS, diagnostic devices fall outside of my area of expertise. Therefore, please take these comments with the appropriate caution that I am a non-expert:

The use of a dried yeast-based detection kit is clever. But is there really an unmet medical need here? What precisely would this be used for? The authors claim this is needed, but give no further explanation. It is not clear to me under exactly what circumstances this would be used medically. As the authors point out, some synthetic cannabinoids such as "Spice"/"K2" are dangerous and one can easily envision why detection of these illicit drugs would be important in an emergency room setting. However, the danger of these drugs comes from structure-based (non-CB receptor-mediated) "off-target" effects of these drugs; and therefore a more traditional assay that detects the actual structures, rather than the amount of CB1 activation, would be required. The CB1 activation itself is not where the danger lies. Perhaps the real use for this device would be in law enforcement, which is much more difficult to get excited about. Furthermore, kits capable of detecting illicit cannabinoids from body fluids within minutes (as opposed to the 3 hour timeframe of this assay) already exist. Also, the authors have not shown any kind of data to speak to the

consistency and reproducibility of this detection technology, which would be very important for either medical or law enforcement applications. Is there day-to-day "drift" in the agonist potency, as there is with many cell-based assays? How does it change under different conditions (temperature, etc.)? How stable is the yeast and the assay over time? How must it be stored? What is the rate of false positives or false negatives?

The authors make much out of the fact that the final detection step can be done on a cell phone – but why? The assay takes hours to incubate, it uses special media (which requires refrigeration) to activate the yeast... Will a cell phone be used to actually analyze the data? Repeating over and over that it can be read using a cell phone feels like it is meant to imply a level of simplicity, portability and convenience that is not really there. Overall and in general, the manuscript reads a bit too much like part of a pitch for potential investors. Figure 1A is one example which doesn't seem to add much scientifically. Similarly, the section of Table 1 with the +s seems relatively meaningless.

Specific details:

1) Assays that display bell-shaped concentration-response curves are problematic for HTS, and would certainly be problematic for medical monitoring. At least three compounds (11-OH-THC, dugesialactone, and "Agonist 2") display bell-shaped concentration-response curves in this assay. The authors do not address this except with one sentence: "Interestingly, at concentrations above 10 μ M, DL acts as an inverse agonist." If this compound really were an agonist at low concentrations and an inverse agonist at higher concentrations, this would be a truly amazing and unique compound and it would be the most interesting finding of this paper. That's not what's going on here; this is most likely some kind of assay interference that is not CB receptor-mediated.

2) Why do the authors use CP55940, instead of THC, to validate their assay and detection kit? How often are people going to be using their kit to try to detect CP55940? It feels like the authors are "cheating" by using the less relevant compound, but one that produces a much bigger signal in their assay.

3) Signal/noise ratio (SNR) is not a good way to quantify the robustness of an HTS assay, for reasons that are illustrated well in Figure 5. The luciferase assay shows a dramatically larger SNR (5C), but inspection of Figure 5B readily shows that this only reflects a slightly higher (but still near-zero) background signal for both of the other two assays. The accepted way to quantify HTS assay robustness is the use of a z' score. This methodology is actually described pretty well on Wikipedia: <https://en.wikipedia.org/wiki/Z-factor>

In summary, the authors have generated a yeast-based GPCR screening assay and shown that it can be used for screening. Overall, the assay development is scientifically sound. However, in my opinion, the importance and likely impact of this work is exaggerated. With some edits, this paper would be appropriate for other journals, such as ASSAY and Drug Development Technology, or Journal of Biomolecular Screening; but I do not see it as impactful enough to warrant publication in Nat Comm, which I see as a very prestigious journal.

Reviewer #3:

Remarks to the Author:

The authors took advantage of the modular GPCR sensor and engineered a CBD sensor with improved sensitivity, signal noise ratio and dynamic response range. The authors also demonstrated the utility of these sensors to screen agonist or antagonist for the CBD receptor. A colorimetric or luminescence reporter system was also established to demonstrate the portability of the sensor. I recommend the publication of this manuscript with major revision.

(1) Line 133-134, does the author mean "Fig 2" instead of "Fig. 3"? It seems that the initial biosensor takes exactly the same genetic configurations or components from Shaw and co-workers (Ref 5). Please explain the difference of this work versus Shaw's work.

(2) After the MFaSS modification, the LOD (level of detection) of both sensor is reduced. Does that mean the total number of receptors on the membrane is also decreased?

(3) After MFASS modification, the EC₅₀ (binding affinity) of the sensor was improved. Please explain the biophysical or structural basis why the EC₅₀ becomes smaller (or bind more tight between the ligand and the receptor).

(4) The portability of the sensor was demonstrated by using the colorimetric reaction of DOPA-5-glucosyltransferase chemistry. Please comment how fast the cell will develop the color. Upon mixing the cell with the CBD analogs, the duration of color development stage is also an important factor to evaluate the sensor performance. If it takes too long, this won't be a rapid sensor.

(5) For the luciferase reporter system, what happens if there is no access of a luminescence detector? The portability will depend on many things.

(6) The direct detection of saliva sample is a big achievement of this paper. Saliva or urine samples may have lots of variations even for the same patient. Please comment how to reduce the detection error. Could this system be used to detect human blood serum?

(6) THS is reported to have strong psychoactive effect. Can the sensors developed in this work be used to distinguish structurally similar trans- Δ^9 -THC, CBD, CBC, and CBG, CBDA and CBGA? For drug screening or illegal THC detection, the detection of full panel of CBD analogs will be more important. Please explain how to improve the specificity of the GPCR sensors.

NCOMMS-21-28728
Response to reviewers (10 Apr 2022)

Dear reviewers,

Thank you for your considerable effort in reviewing this manuscript. We find the comments highly relevant and feel resolving each of them has helped us both in developing our research and improving this manuscript. In order to address each of the comments individually, we have performed more than 15 different additional experiments, added new figures, and revised the manuscript text accordingly.

Changes in the text and figure legends are highlighted in red font. Line numbers refer to the annotated version of the revised manuscript.

Reviewer #1:

Comment 1.1: The innovative idea is to clone human or humanized GPCR CB2 into yeast to mimic G coupled receptor signal transduction into yeast. The first question here, why did the authors choose CB2 and not CB1?

Response: CB2 was chosen because of two main reasons:

1. CB2 is able to detect both natural cannabinoids from cannabis plants and synthetic cannabinoids found as street drugs (all documented such drugs so far are also CB2 agonists).
2. CB2 has an untapped potential as a therapeutic target. Specifically, compared to CB1, CB2-based therapies are vastly understudied. According to current knowledge, CB2 is primarily expressed in active inflammation. Thus, synthetic CB2 ligands show promising results as anti-inflammatory and immunomodulatory drugs. CB2 based therapies are especially relevant in diseases where there are limited options for treatment, such as Alzheimer's, MS or arthritis. However, although many CB2 ligands are in clinical trials, no approved CB2 drug yet exists. Thus, discovering new CB2 specific drug leads is still very relevant.

Therefore, we felt that basing our proof-of-concept study on a receptor that can also serve as a prime pharmacological target has considerable added-value. The text was amended **in lines 40-45 of the annotated revised manuscript** to clarify the motivation behind selecting CB2.

Comment 1.2: The system is used as a biosensor system to detect CB2 mimicking compounds or drugs. The system is built up in a modular way where Gbeta and Ggamma have not been protein engineered. The detection unit is coupled to various actuator modules and all have been discussed in detail. The

work is significant, because it gives a new insight in a bioassay detector system. This may provide a new approach for High Throughput Screening. Advantage of the system is the flexibility by design to adapt various GPCR like structures for various potential ligands. This is an important feature because of allosteric modulation, but that has not been addressed in this manuscript. Here, the authors must discuss limitations of the system due to many allosteric sites in GPCR have an impact on signal transduction.

Response: Thank you for highlighting the flexibility of our system and its strong potential for biodiscovery. The point made here about allosteric modulation has two components.

1. One component is the ability of the biosensor to identify allosteric modulators of CB2. Such molecules could be very interesting as drug leads. As shown in the first proof-of-concept application, our biosensor is indeed very well suited for screening chemical libraries for such compounds, as it is able to identify several molecules that act as CB2 antagonists by modulating the response of the biosensor to a full agonist (**lines 410-411**). When the biosensor is applied in this configuration, it can efficiently identify compounds that modulate the CB2 response. Due to the high dynamic range of the luminescence reporter, the same configuration can subsequently be used to characterize the allosteric ligand.
2. The second component is the susceptibility of the biosensor to interference by the presence of allosteric modulators of CB2 in the samples. This is an important point that we have taken into consideration when designing the case-studies reported here, and should also be considered when validating the biosensor for a new application.
 - a. Among the applications described in this work, bioprospecting is probably the most likely to be affected by allosteric modulators. In this case, efficient fractionation of the plant extract can help separate potential allosteric modulators from possible CB2 orthosteric ligands. In our workflow, fractionation takes place immediately after the initial identification of an interesting plant extract, limiting the risk in subsequent steps. Complementing the bioprospecting set-up with a parallel “antagonist” screening configuration (as the one described in **lines 410-411** of the manuscript) can identify possible interference from allosteric modulators in the crude samples or fractions. Further expanding this setup with the control strain (KM207) that has all the same components as the CB2 based biosensor can provide additional information for the presence of ligands that interfere with the performance of the biosensor.
 - b. High-throughput screening of chemical compound libraries is less likely to be affected by allosteric modulation as the samples only include relatively pure compounds in a solvent.
 - c. In the application where the biosensor is used for the analysis of real-life samples, the specific biosensor protocol has to be validated separately with each new type of sample. In the applications described here, we are using the control strain (KM207; see above), which was especially designed to detect non-CB2 specific activities towards any biosensor component (including

effects of allosteric modulators). The text was amended **in lines 333-338** to specifically address this.

Comment 1.3: It is not clear where the GPCR CB2 is located? What organelle, what spatial resolution in the yeast cell? Here, more data like Ab immunofluorescence detection is needed.

Response: Both in mammalian and yeast cells, GPCRs typically localize to the plasma membrane. Thus, to be functional when produced in yeast, mammalian GPCRs must translocate to the plasma membrane. In this work, to improve membrane localization and consequently GPCR biosensor performance, we fused the CB2 receptor with the mating factor alpha signal peptide (**Figure 2**). To show that fusion of the signal peptide improves localization of CB2, we performed two **additional experiments**.

First, we used fluorescence microscopy to detect the localization of a GFP-tagged fusion of CB2. As shown in **Supplementary Figure 4**, fusion of the MFalpha to CB2-GFP clearly improves its localization to the plasma membrane.

Second, we used differential centrifugation to obtain enriched plasma membranes of cells producing a C-terminally 3xHA tagged form of CB2, which we then analyzed by Western blotting. **Supplementary Figure 5** shows a clearly improved signal in the membrane fraction of MFalpha-CB2-3xHA cells compared with CB2-3xHA cells, suggesting that more receptors are present on the membrane when CB2 is fused to the signal peptide.

Comment 1.4: Cannabinoids show very low solubility and accumulate by more than 50% into membranes. Does this have an impact on the bioassay?

Response: To address this comment, we performed an **additional experiment** assessing the amount of ligand that is bound to cells and removed from the solution during an assay. To this end, we incubated the cells with a high concentration of cannabinoid (THC) and subsequently collected the cells and analyzed the levels of cannabinoids by UPLC-MS. Compared to the added amount of THC, we detected negligible amounts of THC in the membrane extract (**Supplementary Figure 14**) and concluded that absorption of cannabinoids in the yeast cell membrane is very low.

Comment 1.5: Are the determined EC50 correct, if most of the drug does not enter intracellular GPCR CB2 receptors?

Response: As mentioned above, the main application of the developed GPCR-based biosensor is the measurement of extracellular ligands and the CB2 receptor must translocate to the plasma membrane to be functional. Thus, the ligand that is present in the assay media is expected to be accessible to the receptor.

Comment 1.6: Why do we see totally, different kinetic data for induced luminescence in Fig. 4?

Response: The dose-response curves for whole cell-biosensors, such as the one from this work, are typically sigmoid-shaped. However, particular ligands and conditions may result in bell-shaped dose response curves (such as Fig 4d of the previous version of the manuscript). Possible reasons for this behavior include allosteric modulation of the receptor through an additional binding site or non-receptor-specific effects on the biosensor machinery. Any of these are likely to occur at very high concentrations.

To address this, we used the KM207 control strain. We found that the bell shaped curve observed in the case of 11-OH-THC results from a negative, non-specific effect of 11-OH-THC on the biosensor (**Figure 4e**). For this reason, in the revised version of the manuscript, we chose to limit the range of ligands used for dose-response curves to the concentrations that we find relevant for cannabinoid detection.

Comment 1.7: If this is an CB2 receptor, why did the authors not choose CBD or another CB2 specific ligand?

Response: Thank you for this suggestion. In the revised manuscript, we have now included CBD among the ligands tested with our biosensor. In **Figure 4h**, we now show the result of CBD determination using the biosensor.

Comment 1.8: This brings up the question, if we can distinguish between a full an partial agonist / antagonist?

Response: Yes. It is possible to distinguish between a full and a partial agonist. Partial/full agonism of a ligand can be determined by comparing the maximum output (the output obtained with a saturating amount of the ligand) with the maximum output obtained with a known full agonist, such as CP55940. A partial agonist will show considerably lower maximum output. As an example, in this work, we show that incubating the biosensor with THC (a partial agonist) results in less than 50% of the max output with (the full agonist) CP55940 (**Figures 2a and 2b**).

Comment 1.9: I like the idea of a portable CB2 detector and recording data with a camera, but I am not sure if this makes sense to get valid data out of a crude mixture of a biological sample.

Response: We provide validation of the function of the portable biosensor in complex biological samples by testing it in three different (artificial) human bodily fluids, saliva, urine and serum (**Figure 8g**).

Furthermore, we show that the biosensor can also be functional in the most challenging matrix, such as the complex natural chemical extracts encountered during bioprospecting (Figure 6b).

Comment 1.10: Overall an interesting report with a new concept of using heterologous GPCR CB2 receptors. But, we should be aware that cannabinoid-like effects are orchestrated by CB1, GPR55 and COX2 in parallel. Therefore, a yeast based system will come fast to its limitations.

Response: Our system will indeed have limitations in predicting the total physiological effects of known and putative cannabinoids. Cell-based assays are also only partially able to answer this question, which ultimately requires validation in animals. However, it has never been our goal to utilize the yeast system to fully characterize the action of CB2-targeting ligands.

Our goal is to provide a fast, economical and complementary technology that has the added advantage of orthogonality. Our method can be used to study CB2 and its direct ligands in relative isolation. Frequently, cell-based GPCR studies are challenged by off-target effects of the studied ligands and using an orthogonal, yet sensitive, robust and economical, assay such as the one presented here, could be highly beneficial. Selected candidates can be further characterized in cell-based and animal models.

Comment 1.11: The conclusions drawn are fine, but there is still much space for improvement. TO give one example, the EC50 range is in the micromolar range what is for a new modern drug far too high. Here, the nanomolar range must be the future target.

Response: The determined EC50s are dependent on the affinity of the ligand for the receptor and not limited by the function of biosensor. This is clearly shown for CP55940, where the EC50 determined with the biosensor agrees with the experimentally determined K_D (down to 1.5 nM). Therefore, the biosensor is fully capable of identifying ligands in the nanomolar range.

Comment 1.12: The methodology sounds well and the Danish group is known for solid work in the field of analytics.

Thank you!

Reviewer #2:

Comment 2.1: This manuscript describes the development of a yeast-based reporter gene system for measuring the activity of recombinantly expressed GPCRs. While this system could presumably be used for any recombinantly expressed GPCRs, CB1 and CB2 cannabinoid receptors were used for the current

work. This work builds on previous work that has defined GPCR pathways in yeast, which the authors cite appropriately. The authors optimize this assay system by testing several reporter gene outputs (luciferase is found to be best), and use it for screening both a 1600 compound library of small molecules and 71 samples of natural product extracts vs CB2. The assay is shown to be usable for screening, with active compounds identified from the screening. Importantly, evidence was provided that these hits were "on target" using a "null assay" (same assay but without the recombinant CB2 receptor). Finally, the authors use this assay technology to develop a portable detection kit for detecting cannabinoids from body fluid samples. An advantage of this kit over current cannabinoid detection technologies is that it can detect any kind of CB1 agonist based upon its cellular activity, and therefore is not limited to identifying compounds with known cannabinoid structures.

It was difficult for me to identify precisely where the major scientific advancement is meant to be. Cell-based reporter gene assays are certainly not new. Typically they are not favored for high-throughput screening (HTS), as they are prone to a high rate of false positives because there are many points downstream of the target GPCR where compounds could act to modulate expression of the reporter gene. Therefore, other cell-based assays that measure more proximal events (changes in second messenger levels such as Ca²⁺ or cAMP) are more often employed for GPCR HTS. Reporter gene assays (such as the mammalian PrestoTango assay system) are more typically reserved for "problematic" GPCRs, such as orphan GPCRs where the relevant second messengers are unknown.

Response: As the reviewer points out, current cell-based GPCR assays suffer from "a high rate of false positives because there are many points downstream of the target GPCR where compounds could act". This is exactly where a cell-based system that is orthogonal, like the one presented here, could provide a very useful complementary tool to existing technologies and methods.

The yeast-based system is considerably less prone to off-target effects than mammalian cell-based systems. Yeast provides a minimal cellular chassis that, unlike mammalian cells, does not contain 1. multiple non-target receptors or 2. elaborate downstream signaling pathways, which are major contributors to observing off-targets effects. The problem of multiple receptors being present and the possibility that these receptors contribute to non-specific signaling cannot be solved by measuring more proximal events such as second messengers. By contrast, yeast cells have only one more GPCR in addition to the one studied. Therefore, there is little chance to activate other GPCRs. Importantly, yeast cells are also known to efficiently pump out xenobiotics (Barabote et al., 2011 *Adv Enzymol Relat Areas Mol Biol.* 2011; 77: 237–306., Almeida et al., 2021 *mBio* 21;12(6):e0322121). This feature is highly beneficial in this application because it helps to keep ligands outside of the cells and avoid inadvertent intracellular off-target effects. In addition to these design advantages, we have equipped our system with an appropriate control strain to monitor false positives resulting from ligands acting on the biosensor machinery. In this control strain, KM207, CB2 has been replaced with the A2A receptor.

To add further support to the claim that the yeast system is not prone to false positives, we carried out **additional experiments** where we evaluated all identified agonists and antagonists with the control strain (AM207). These experiments confirmed that the identified compounds

were indeed not false positives. The results of this analysis are now included in the revised figures (**Figures 4b-k, 5c-d, 5f-g, and 6d**).

As the reviewer suggests, there is no perfect GPCR assay, and each of the current systems has advantages and drawbacks. This microbial system is not intended to replace mammalian systems but rather to serve as a convenient and inexpensive first line alternative for the evaluation of novel compounds, particularly in early drug discovery.

Comment 2.2: For researchers who are experienced with reporter gene assays, it will certainly come as no surprise that the luciferase version led to the best assay compared to a fluorescent protein or colorimetric readout.

Response: In this work, we compare three different biosensor strains to find which one is best suited for each of the different applications. HTS was only one of these applications. Although the luciferase reporter was found to be preferable for HTS applications, we found that for other applications a different reporter was preferable. For example, the colorimetric reporter strain enables assays that can be read by naked eye and has lower cost. Furthermore, the fluorescent reporter has the benefit that its quantification does not require extra manipulations, such as substrate addition or cell lysis, thus allowing continuous measurement.

Comment 2.3: The researchers claim that their assay can “democratize drug discovery by enabling low-cost high-throughput analysis using open-source automation.” Yes, open-source automation has significantly reduced the cost of setting up an HTS laboratory; but that is completely independent of this paper. This work does not enable or expand the impact of this cheaper lab automation in any way; this assay benefits from the less expensive equipment the same way that any assay would. Although the authors claim throughout the paper that their assay provides a “far more economical and user-friendly format” for HTS, there is no explanation of how exactly. Likely the authors are referring to the fact that yeast are somewhat less expensive to grow and maintain than mammalian cells, but this represents marginal savings in the overall big picture of an HTS campaign; and the HTS campaign itself (typically \$50-100K, depending upon size of the library and other factors) is a relatively insignificant portion of the \$10-15M it takes to bring a new compound into the first phase of clinical trials. This assay will hardly democratize drug discovery.

Response: In this work, we demonstrate proof-of-concept of a simple platform for testing individual compounds, complex natural extracts, or chemical libraries using minimal resources and no specific expertise in mammalian cell assays. We envision that the existence of such a platform will lower the threshold for numerous researchers to test their novel compounds or extracts in their labs. Thus, drug discovery will be available to a much wider range of academic and small-scale commercial labs. Because our strains are freely available to the scientific community, the equipment and reagents required for the analysis are economical, and there is the potential of further co-development of strains for different GPCRs by the community

(which will then again be broadly available), our technology will make the discovery of GPCR ligands available to many more labs. This is the essence of the argument about “democratization of drug discovery”.

We acknowledge that the way the sentence in the abstract was written was misleading as it included the connection to open-source automation (which is a facilitator to the above but not the enabling factor). This was not our intention and this sentence has now been rephrased in the abstract and the main text (**abstract and lines 377-390 and 596-607**).

Furthermore, we would like to comment that the costs of drug screening quoted by the reviewer do not offer a direct comparison with our system. The budget of a campaign that extends to \$10-15M includes mostly the cost of different activities, such as safety and clinical trials. Here, we believe comparison should be made at the same level, i.e. the cost of screening itself. Comparing the cost of using the cell line to the cost of the entire process from drug discovery to clinical trials is hardly relevant. Neither, the 50-100k USD mentioned by the reviewer, that likely represents the price of the library itself and reagents, is a comparable cost. However, the costs of setting up mammalian cell-based assay facilities and employing dedicated personnel can be prohibitive for a non-specialist lab.

Comment 2.4a: Furthermore, it has significant disadvantages: 1) a great deal of genetic engineering of the yeast is required, which involves its own money and time cost.

Response: This is not true. On the contrary. The biosensor described in this study is based on a general platform strain (chassis) that can in principle be used to express any GPCR. This optimized platform strain, KM111, has been developed in this work and will be available to the entire research community. To study another GPCR, the interested lab will only need to integrate the receptor and the corresponding Galpha. Compared with establishing a novel mammalian reporter line, inserting these two genes using the highly efficient yeast genetic engineering tools available is far more convenient and economical. Furthermore, mammalian cell systems are often based on transfected cells that have to be prepared anew for use in different labs or cryopreserved. On the contrary, once a yeast biosensor strains has been established, this strain is stable, requires simple storage, and can readily be exchanged among different labs as a tool.

Comment 2.4b: 2) It utilizes a yeast-based cell system, rather than mammalian, taking the GPCR target one step further away from its native, physiologically relevant environment.

Response: We propose that using a yeast system employing an orthogonal sensing pathway may offer a complementary solution to mammalian cell-based systems. One of its main advantages arises from its orthogonal setup. In this sense, taking the GPCR away from its native environment is not a shortcoming.

Comment 2.4c: 3) Physiological relevance (at the molecular level) is further reduced by the fact that a) the yeast's natural G protein regulator protein (SST2) is knocked out to increase signaling,

Response: Since we are constructing an orthogonal yeast system, we do not aim to accurately reproduce the amplitude and duration of downstream signaling exhibited by human cells. In the context of an orthogonal biosensor, removal of the regulatory function of SST2 is not a shortcoming but rather an advantage. The function of GTPase accelerating proteins, like SST2, is to regulate the amplitude and duration of the effect of GPCR activation. As a result, a mutant strain lacking SST2 is likely to demonstrate more pronounced effects of GPCR activation. The likely implication for an HTS agonist screen is expected to be somewhat linear amplification of the output resulting from hits. This can be an advantage, as it improves the sensitivity and can help uncover more subtle effects of some ligands. We expect relevant hits to still stand out (as can be observed from **Figure 5b**). Furthermore, in the case of antagonist screening, greater signal amplitude is expected to result in better resolution of antagonist effects.

As an additional note, we would like to point to the fact that the physiological relevance of the amplitude and duration of downstream signaling from mammalian tumor-derived cell lines overexpressing GPCRs is also questionable.

Comment 2.4d: b) the G protein used is a chimeric yeast/mammalian mutant G protein,

Response: We agree that using a yeast/mammalian chimeric Galpha may result in different CB2 activation behavior for some compounds. However, we would expect a partial effect rather than complete inactivation. Considering the high SNR of our CB2 biosensor, we expect that it will still be able to detect the specific ligand.

As human GPCRs have been shown to also couple (albeit less efficiently) with wild-type human Galpha proteins in yeast, in the future the biosensor could be improved by developing a system based on a human Galpha. In this case, the yeast Gbeta and Ggamma would be engineered to perfectly couple with the human Galpha.

Comment 2.4e: and c) the GPCR itself is artificially fused to another (yeast) protein, with unknown consequences for receptor 3-D structure and function.

Response: MFaSS contains a cleavage site that is shown to promote efficient cleavage of the signal peptide. We confirm this in **an additional experiment (Supplementary Figure 5)**, in which we demonstrate that the 9 kDa MFaSS tag is fully cleaved from CB2. Thus, we expect that this will not have an effect on the structure or function. Receptor protein preparations produced for radio ligand assays and crystallography of GPCRs have often been produced in yeast using the exact same signal peptide (Bertheleme et al., 2015 *Methods Enzymol* 556:141-64).

Comment 2.5: Whatever the authors' view about the recent trend toward more physiologically relevant screening assays may be (I personally feel that this is in many cases taken too far), we all must recognize that this trend does exist. Even mammalian cell-based assays that utilize a mutant G protein in order to couple GPCR signaling to a more convenient readout (use of Gqi5 mutant G protein to artificially couple GPCRs to Ca²⁺, for instance) have largely fallen out of favor as drug discovery researchers have begun putting greater and greater emphasis on physiological relevance, even for the assays used for HTS.

Response: We acknowledge the advantages of performing GPCR assays under conditions as close to natural as possible. However, our aim was never to compete with such platforms. We see our system as a simple, orthogonal and economical option for first line screening, which can function complementary to more physiologically relevant (but more demanding and expensive) assays, and we believe that it will be well received and adopted as such.

Comment 2.6: In short, I see a number of potential disadvantages to using this or a similar assay system for HTS, with the only potential advantage being a relatively small cost savings/increase in convenience that could come from using yeast vs mammalian cells. I honestly don't see this assay getting much use as an HTS assay.

Response: Possibly, traditional established actors in the pharma industry would not immediately shift to using a yeast biosensor for HTS. However, we envision that many academic laboratories, non-established companies, start-ups, innovators, etc would be interested in a system that can be easily set up and operated with low cost. Using state-of-the-art equipment and cell lines is a luxury that not all can afford and it keeps many labs from testing their compounds and proteins.

Yet, this technology could have an indirect impact for pharma. As reviewer #2 points out, screening is only a part of drug development. Enabling many smaller actors, like for example chemical biology or natural product chemistry labs, to carry out efficiently the early steps of drug discovery could bring benefits similar to "crowdsourcing" for larger companies that are able to commercialize these new leads.

Comment 2.7: Perhaps the real novelty/innovation of this manuscript is meant to lie in the portable cannabinoid detection device that the authors have created. I must admit that while I am an expert in HTS, diagnostic devices fall outside of my area of expertise. Therefore, please take these comments with the appropriate caution that I am a non-expert: The use of a dried yeast-based detection kit is clever. But is there really an unmet medical need here? What precisely would this be used for? The authors claim this is needed, but give no further explanation. It is not clear to me under exactly what circumstances this would be used medically.

Response: The cannabinoid biosensor itself has many different potential uses for example in medical diagnostics and research, cannabis breeding, and law enforcement. In the context of medical applications, the sensor can be used to detect the levels of cannabinoids in bodily fluids such as urine, saliva or serum. Endocannabinoid levels in serum can be used as biomarkers for diverse conditions (Kratz et al., 2021 J Mass Spectrom Adv Clin Lab. 22: 56–63), which could be monitored using this biosensor. Furthermore, it could be used to monitor the efficiency of the treatment of cannabinoid-treated (MS or other disease) patients. This can be done in a practitioner’s office, without the use of dedicated equipment or sending samples to a hospital lab. Outside the medical context, the cannabinoid biosensor can be used for analyzing large numbers of samples in cannabis breeding programs (**lines 631-633**). As discussed by the reviewer (in comment 2.8 below), it can also be used for the detection of (current or novel) illicit synthetic cannabinoids for which other tests do not exist (**lines 631-633**).

But the most important aspect of our work is that it shows that it is possible to construct easy-to-use whole-cell biosensors employing GPCRs for use outside the lab by non-professionals. While we demonstrate this using CB2 to detect cannabinoids, different GPCRs could be utilized according to the same portable workflow to detect many other compounds. We envision that the system can easily be adapted to detect numerous other important ligands including disease biomarkers, drugs, hormones, pesticides, pollutants and other substances from both human and field-collected samples.

Comment 2.8: As the authors point out, some synthetic cannabinoids such as “Spice”/”K2” are dangerous and one can easily envision why detection of these illicit drugs would be important in an emergency room setting. However, the danger of these drugs comes from structure-based (non-CB receptor-mediated) “off-target” effects of these drugs; and therefore a more traditional assay that detects the actual structures, rather than the amount of CB1 activation, would be required. The CB1 activation itself is not where the danger lies.

Response: The advantage of our biosensor is that it can detect compounds based on their ability to bind the receptor, therefore it is not limited by the structure of the ligand. Its advantage is particularly evident in the case of synthetic cannabinoids, because it can detect in one assay all possible compounds, known and unknown, current or future, irrespective of their structure. All synthetic cannabinoids by definition target the cannabinoid receptors, irrespective whether their dangerous effects are due to CB1-overstimulation or off-target effects. In this context, more traditional assays that detect actual structures are not a comparable solution, while our biosensor will also be able to detect future synthetic cannabinoids emerging on the street market.

Comment 2.9: Perhaps the real use for this device would be in law enforcement, which is much more difficult to get excited about. Furthermore, kits capable of detecting illicit cannabinoids from body fluids within minutes (as opposed to the 3 hour timeframe of this assay) already exist.

Response: As discussed above, the advantage here comes from the fact that beside THC (or 11-COOH-THC) and CBD, there are no available quicktest kits for many current or future illicit cannabinoids. This is especially important for detection of synthetic cannabinoids (designer drugs), since new compounds with different structures emerge on the market on a regular basis. A biosensor capable of detecting such compounds regardless of their structure would be very useful, particularly because with continuous introduction of new designer drugs the development and use of individual kits for each compound detected would be required. For example the biosensor could be used for intercepting such compounds at customs.

Moreover, in the experiments presented here, we mostly used an assay time of 3 h for convenience because this would ensure maximal signal for the luminometric biosensor strain. However, with the same detection settings this biosensor is capable of a signal to noise ratio of >25:1 already after 30 min (**shown in Supplementary Figure 7**). We also show that time response is similar regardless of compound (**Supplementary Figure 13**). Since maximum output with the partial agonist THC is typically approx 40% of the maximum output (with the full agonist CP55940) (**Figure 4**), we expect to be able to confidently detect THC and similar potency cannabinoids in 30 min.

In principle, cannabinoids could be detected in even less time. In this study, we focus on optimizing the biosensor strain itself. However, there is a lot of scope for technical improvements in the biosensor device (more focusing optics, shorter measuring distance) and cell phone software (more powerful signal processing) that could reduce detection time (discussed in **lines 634-637**).

We would like to take the opportunity here to remind once more that the main aim of this work is to provide proof-of-concept that GPCRs can be used to establish whole-cell biosensors with robustness and sensitivity suitable for real-life applications. As such, the examples provided here are not meant to already be blockbuster products in the market but to demonstrate that the technology can provide efficient and trustworthy solutions.

Comment 2.10: Also, the authors have not shown any kind of data to speak to the consistency and reproducibility of this detection technology, which would be very important for either medical or law enforcement applications. Is there day-to-day “drift” in the agonist potency, as there is with many cell-based assays? How does it change under different conditions (temperature, etc.)? How stable is the yeast and the assay over time? How must it be stored?

Response: We addressed this comment with additional experiments and found that our biosensor produces results with considerable reproducibility. When comparing three measurements with strain KM206 made in three different days, the standard deviation (“day-to-day drift”) in EC50 was 5.6% (shown in **Supplementary Figure 13**).

Temperature is known to affect the binding affinity of GPCRs for different ligands (K_D). Thus, temperature will also have an effect on the biosensor’s behavior. In an **additional experiment**, we determined the effect of the temperature on the EC50 for CP55940 (**Supplementary Figure**

15). In the range from 20 °C to 37 °C, the EC50 for CP55940 changes progressively from 0.82 nM to 2.27 nM, following a trend and rate (approx. 7%/°C) that are in good agreement with the theoretical effect of temperature on the equilibrium binding constant.

Furthermore, temperature is also expected to have an effect on the different reporters employed by the biosensor strains. Therefore, we performed an **additional experiment** to examine the effect of temperature on the maximum output of the different biosensor strains. All three reporter strains were functional at all temperatures tested. With all of them the maximum output was observed at 25 °C (**Supplementary Figure 6**). This is advantageous, as the portable biosensor is expected to be most frequently used at room temperature.

These temperature-dependent changes in EC50 and maximum output are not expected to have a considerable impact on the useability of the biosensor. First, and most importantly, because the (portable) biosensor contains an integrated calibration system based on cannabinoid standards, which allows it to produce consistent results irrespective of temperature fluctuations. Second, because the biosensor is expected to be typically used in a relatively narrow temperature range (around room temperature).

Regarding the question as to how the biosensor is supposed to be stored, dried yeast is known to stay viable and active for long times. We performed an **additional experiment (Supplementary Table 7)** demonstrating that our biosensor maintains 92% of its activity (maximum signal) for at least 1 month at 4 °C when preserved in 0.1 M potassium phosphate (pH 7), 0.1% ascorbic acid, bubbled with N₂ and sealed. However, we expect the biosensor to stay active considerably longer than that, since a similar yeast-based biosensor has been shown to retain at least 50% of its activity for up to 38 weeks at room temperature (Ostrov et al., 2017 Sci Adv 28;3(6):e1603221)). We expect that this small decrease of activity over time does not pose a considerable problem, since, as mentioned in the previous paragraph, the portable biosensor protocol includes comparison of the readout of the sample to simultaneously obtained readouts of cannabinoid standards of different concentrations (**Figure 8f**).

Comment 2.11: *What is the rate of false positives or false negatives?*

Response: In the case of the portable biosensor, in the experiments carried out in the context of this manuscript, we did not observe any false positives or false negatives. However, this is a valid concern when it comes to the analysis of human samples and, for this reason, we have designed the portable biosensor workflow to be able to detect false positives or negatives. Specifically:

1. We have introduced calibration wells with known amounts of a cannabinoid that act as positive controls and enable detection of false negatives resulting from a non-functional biosensor.

2. To detect a false negative effect on the biosensor coming from the sample matrix, the sample can be applied on a cannabinoid-containing well and the output compared to that of the corresponding matrix-free calibration point and the actual measurement.
3. Identifying false positives resulting from non-CB2 specific activation effects of the sample matrix can be done by introducing a negative control containing the non-CB2 control strain (A_{2A} strain) as part of the workflow.

Comment 2.12: The authors make much out of the fact that the final detection step can be done on a cell phone – but why? The assay takes hours to incubate, it uses special media (which requires refrigeration) to activate the yeast... Will a cell phone be used to actually analyze the data? Repeating over and over that it can be read using a cell phone feels like it is meant to imply a level of simplicity, portability and convenience that is not really there.

Response: We kindly disagree. Mobile (smart) phones have developed into portable computers that have powerful cameras and processors capable of elaborate data processing and analysis. Employing such devices in connection with biosensors enables analysis outside the lab without specialized equipment. In fact, we envision that mobile (phone) applications will gradually become commonplace in portable analytics in multiple fields.

In the context of the portable biosensor, the use of a cell phone camera and software can considerably decrease the time required for analysis because the software can continuously sample the light produced and analyze the measurements in real-time and control the duration of the sampling/analysis until a confident result is obtained. Moreover, the software can compare the positive and negative controls and also perform semi-quantitative calculations using the calibration curve.

Regarding the biosensor response time: In this work we mostly use 3 h incubation time for consistency and to obtain the strongest possible signal. However, as mentioned earlier, we also show that the biosensor can readily detect cannabinoids after 30 min (and probably less). Furthermore, we are confident that the response time can still be considerably improved with further technical development of non-biosensor parts. Specifically, improvements to the biosensor hardware and software such as the use of condensing optics and advanced signal processing, respectively, could dramatically increase the biosensor range and response time.

Moreover, there may be a misunderstanding by the reviewer because the reagents used in the biosensor assay do not require refrigeration. Both media and luciferase assay components can be stored in dried form and reconstituted just before the assay. This was not exploited further here, because in this work we concentrated on the biosensor itself and further product development was outside our scope.

Comment 2.13: Overall and in general, the manuscript reads a bit too much like part of a pitch for potential investors. Figure 1A is one example which doesn't seem to add much scientifically. Similarly, the section of Table 1 with the +'s seems relatively meaningless.

Response: This work introduces a novel concept of portable whole-cell biosensors and is thus focused on future possibilities brought about by such devices. However, we want to avoid the impression of over-selling. Thus, parts of the manuscript were re-written to address this.

Figure 1A aims to convey the message that the biosensor is suited for multiple different types of applications each having different requirements. In the same vein, Table 1 (in the previous version) was used to compare the different biosensor strains to display their suitability for different applications. However, in the new version of the manuscript we decided to shift the focus away from comparing the strains and so decided to omit Table 1.

Specific details:

Comment 2.14: 1) Assays that display bell-shaped concentration-response curves are problematic for HTS, and would certainly be problematic for medical monitoring. At least three compounds (11-OH-THC, dugesialactone, and "Agonist 2") display bell-shaped concentration-response curves in this assay. The authors do not address this except with one sentence: "Interestingly, at concentrations above 10 uM, DL acts as an inverse agonist." If this compound really were an agonist at low concentrations and an inverse agonist at higher concentrations, this would be a truly amazing and unique compound and it would be the most interesting finding of this paper. That's not what's going on here; this is most likely some kind of assay interference that is not CB receptor-mediated.

Response: Indeed, when three of the compounds assayed in this work were added at very high, non-physiological concentrations, we obtained bell-shaped dose-response curves. We intentionally included very high concentrations in the analysis to fully evaluate the performance of the biosensor. We do not find this to be unexpected or unique to our biosensor. Causes for such behavior may include colloid formation, toxicity, a negative effect on the downstream signaling pathway, ligand inhibition, or negative allosteric modulation of the receptor by the ligand.

To account for such possible effects of the ligands, we integrated a control biosensor strain in our experiments. The control strain is based on the same chassis but contains the A2A receptor instead of CB2 and is activated by a constant concentration of adenosine. This control strain (KM207) can inform if the ligand studied has non-CB2 mediated effects on the biosensor. We used KM207 in **additional experiments** to study the behavior of each ligand. In these experiments, the A2A-based control strain (KM207) was induced with its ligand (adenosine) together with each of the studied ligands separately. To facilitate interpretation of the results, we have now included in **Figures 4, 5 and 6** the output of the control strain for each of the ligands studied (shown as gray line in the same graph). As shown in **Figure 4e**, 11-OH-THC has a negative effect on non-CB2 biosensor component(s) in concentrations above 10^{-5} M. Agonist 2 (AGO2) was also found to interfere with the assay at concentrations above 10^{-5} M, which however is well above the determined EC_{50} (**Fig. 5d**). Interestingly, we confirmed that the behavior of dugesialactone is not due to nonspecific biosensor inhibition but indeed

suggestive of allosteric modulation at concentrations above 10 μ M (where the control strain is fully functional as shown in **Figure 6d**).

Overall, with the addition of the control strain (KM207) we can account for any negative effects of the identified ligands on the biosensor performance and filter out false-positive antagonists. Inclusion of the KM207 control stain in our workflow considerably increases the robustness and reliability of the findings.

Comment 2.15: 2) Why do the authors use CP55940, instead of THC, to validate their assay and detection kit? How often are people going to be using their kit to try to detect CP55940? It feels like the authors are “cheating” by using the less relevant compound, but one that produces a much bigger signal in their assay.

Response: We used CP55940 initially for all biosensor experiments because of its ability to act as a potent full agonist and for consistency (certainly not for “cheating”). In response to this suggestion by the reviewer, we have now expanded the validation of the portable luminometric biosensor by analyzing several relevant compounds, including THC, 11-OH THC and JWH018. We present additional data that demonstrate the biosensor can also readily detect these ligands. In the revised manuscript, we show the response of the biosensor to these ligands in **Figure. 8**. Furthermore, we validated the ability of the betanin-based biosensor to detect THC and show the data in the revised **Figure 7b and 7c**.

Comment 2.16: 3) Signal/noise ratio (SNR) is not a good way to quantify the robustness of an HTS assay, for reasons that are illustrated well in Figure 5. The luciferase assay shows a dramatically larger SNR (5C), but inspection of Figure 5B readily shows that this only reflects a slightly higher (but still near-zero) background signal for both of the other two assays. The accepted way to quantify HTS assay robustness is the use of a z' score. This methodology is actually described pretty well on Wikipedia: <https://en.wikipedia.org/wiki/Z-factor>

Response: We agree that although the signal to noise ratio is useful for determining the suitability of a specific biosensor strain for certain applications, it is not necessarily a good way to quantify robustness of an HTS assay. Thus, in order to determine the robustness of our high-throughput assay, we performed **additional calculations** to determine the Z' score (Zhang et al., 1999 J Biomol Screen 4, 67-73). For the agonist screen, the Z' score was found to be 0.86, while for the antagonist screen it was 0.61 (**Supplementary Table 2**). These scores put both assays in the “excellent assay” category. This is now discussed in **lines 421-424** of the revised manuscript.

Comment 2.17: In summary, the authors have generated a yeast-based GPCR screening assay and shown that it can be used for screening. Overall, the assay development is scientifically sound. However, in my opinion, the importance and likely impact of this work is exaggerated. With some edits,

this paper would be appropriate for other journals, such as ASSAY and Drug Development Technology, or Journal of Biomolecular Screening; but I do not see it as impactful enough to warrant publication in Nat Comm, which I see as a very prestigious journal.

Response: We strongly feel that this work is perfectly suited for publication in Nature Communications because it makes an important conceptual advance in the field of synthetic biology and is appealing to a broader scientific audience.

This work will have great impact in the field because it will inspire the use of GPCR-based biosensors for numerous other applications and will drive further development of the technology and additional research publications from numerous labs. Providing the design, the strains, and the updated reporters to the community will enable the rapid co-development of GPCR-based biosensors. Numerous labs working in natural products or chemical synthesis will have simple and economical assays in their hands to evaluate their compounds, small companies will be able to develop more economical HTS campaigns, and researchers working on specific GPCRs for which convenient mammalian systems do not exist, will have additional tools. Moreover, the ability of these biosensors to be used in portable, out-of-lab applications will inspire the development of biosensors for numerous other molecules.

Thus, this manuscript clearly fulfills Nature Communications' mandate to publish high-risk/high gain research that opens up new directions.

Reviewer #3:

The authors took advantage of the modular GPCR sensor and engineered a CBD sensor with improved sensitivity, signal noise ratio and dynamic response range. The authors also demonstrated the utility of these sensors to screen agonist or antagonist for the CBD receptor. A colorimetric or luminescent reporter system was also established to demonstrate the portability of the sensor. I recommend the publication of this manuscript with major revision.

Comment 3.1: *Line 133-134, does the author mean "Fig 2" instead of "Fig. 3"?*

Response: Thank you for pointing this out. This was a mistake and has been corrected in the revised text.

Comment 3.2: *It seems that the initial biosensor takes exactly the same genetic configurations or components from Shaw and co-workers (Ref 5). Please explain the difference of this work versus Shaw's work.*

Response: Indeed, our study builds upon the work of Shaw and co-workers (Shaw et al., 2019 Cell 177, 782-796) and introduces additional improvements to achieve optimal performance in

specific applications. These modifications aimed at two primary goals, to improve the sensitivity and to enhance the performance of the reporters.

Briefly, specific modifications aiming to improve sensitivity and dynamic range include: 1. using a different chassis strain that has reduced ER protein degradation, 2. utilizing different integration loci for all pathway components, 3. use of a different promoter (P_RET2 instead of P_RAD27) to drive the master transcription factor gene STE12 in order to achieve improved response, and 4. fusion of the MfaSS to CB2 to improve membrane localization. These modifications are described in detail in **lines 154-157, 156-157, 161-167 and 189-191** of the revised manuscript.

Regarding the reporter module, we optimized the reporters for optimal performance in each of the three selected applications. The betalain reporter, in particular, is novel and is developed here especially for use in this biosensor (This is described in **lines 272-289**). The luminescence reporter on the other hand, is developed to suit a fast portable format and is coupled to a custom-built device so that it can be read by a mobile phone (**Figure 8**).

***Comment 3.3:** After the MFaSS modification, the LOD (level of detection) of both sensor is reduced. Does that mean the total number of receptors on the membrane is also decreased?*

Response: A lower LOD (limit of detection) means that the biosensor is more sensitive. This improved sensitivity is likely due to having more receptors on the membrane as a result of fusing MFaSS to CB2. As explained in the response to comment **1.2**, we have confirmed with additional experiments that when MFaSS is fused to CB2 there are more receptors present in the plasma membrane fraction (**Supplementary Figure 5**), while GFP-tagged MFaSS-CB2 is more efficiently localized at the membrane than CB2-GFP (**Supplementary Figure 4**). This is consistent with the model proposed by Shaw and co-workers, which suggests that the presence of more receptors on the membrane results in higher sensitivity (Shaw *et al.*, 2019 *Cell* 177, 782-796)).

***Comment 3.4:** After MFaSS modification, the EC₅₀ (binding affinity) of the sensor was improved. Please explain the biophysical or structural basis why the EC₅₀ becomes smaller (or binds more tight between the ligand and the receptor).*

Response: The apparent EC₅₀ value describes the overall performance of the whole-cell biosensor and does not strictly reflect the ligand equilibrium binding constant K_D . As described in Shaw *et al.*, other factors that could be limiting under the specific experimental conditions or design may have a considerable influence on the behavior of the biosensor and, thus, the apparent EC₅₀. This has been well covered in Shaw *et al.*, 2019 *Cell* 177, 782-796 and Bush *et al.*, 2016 *Mol Syst Biol* 12, 898. In the case of our CB2 biosensor, when the MFaSS is absent, the performance of the system appears to be limited by other factors, such as the stoichiometry of the alpha or beta/gamma proteins, and the observed EC₅₀ is higher than the measured K_D for CP55940 (Soethoudt *et al.*, 2017 *Nat Commun* 8, 13958). However, when the MFaSS modification is introduced, the performance of the biosensor improves considerably and, as a

result, the observed EC50 reaches the experimentally determined K_D (**lines 197-201 of revised manuscript**). This indicates that under conditions where there is a limited amount of CB2 on the plasma membrane (or improper insertion of CB2 in the membrane), the efficiency by which the binding event is coupled to the reporter output is compromised, resulting in higher apparent EC50.

We do not consider that the presence of the MF α SS peptide is likely to have a direct positive influence on the binding affinity itself, because the MF α SS includes a cleavage site that directs its efficient removal from the receptor upon membrane localization. We can confirm that under the conditions of our experiments, the MF α SS peptide (mass 9 kDa) has been completely removed because this is apparent from the data shown in **Supplementary Figure 5**, where CB2-3xHA and MF α SS-CB2-3xHA have the same mobility in SDS-PAGE.

***Comment 3.4:** The portability of the sensor was demonstrated by using the colorimetric reaction of DOPA-5-glucosyltransferase chemistry. Please comment how fast the cell will develop the color. Upon mixing the cell with the CBD analogs, the duration of color development stage is also an important factor to evaluate the sensor performance. If it takes too long, this won't be a rapid sensor.*

Response: **Supplementary Figure 7** shows a time course of color development of the colorimetric biosensor strain (KM205 + o-Da) with a saturating concentration of CP55940. A color distinguishable by eye can be seen already after 3-4 h. However, the maximum colorimetric signal takes 13 h to develop.

At this stage of development, we do not consider the colorimetric strain a rapid sensor. Rather, we propose that this biosensor is well suited for applications that require the analysis of a very large number of samples in parallel (but where speed is not a requirement). Such a workflow could be used in different mass testing applications, for example quality control of cannabis products or in cannabis breeding. In the future, we will continue developing the speed of this reporter. Possible improvements include, for example, increasing the copy number of the reporter gene (MjDOD).

For applications that require rapid cannabinoid detection, we recommend using the biosensor with luminometric output (KM206). Using this biosensor, we can obtain a signal that is >25-times above the background in 15-30 min at 25 °C.

***Comment 3.5:** For the luciferase reporter system, what happens if there is no access of a luminescence detector? The portability will depend on many things.*

Response: One of the reasons to choose luciferase as a reporter for the biosensor was that it can produce enough light to be sensed with a cell phone camera. Thus, we were able to develop a biosensor that can be used in environments where no lab equipment is available. To facilitate such analyses, in this manuscript, we also construct a simple device that is made from readily

available materials, which can be fitted on most cell phone cameras for easy biosensor reading (shown in **Figure 8b**).

Comment 3.6: The direct detection of saliva sample is a big achievement of this paper. Saliva or urine samples may have lots of variations even for the same patient. Please comment on how to reduce the detection error.

Response: Differences between individual saliva (and urine) samples add a complexity in these types of analysis, which affects, to a varying extent, not only our biosensor but also all other kinds of urine or saliva tests. For example, samples from different individuals may occasionally contain metabolites that interfere with the analysis. The water content and salinity of the sample are other important factors to consider. In our system, we have included relevant controls that can help evaluate the performance of the system under these conditions. For example, we have developed strain KM207 to be used as a control for the correct function of the biosensor components (**manuscript lines 333-339**). The analysis workflow shown in **Figures 7 and 8** can be modified to include one test sample and two control samples employing KM207 with and without adenosine. One additional control that can be introduced to standardize the measurements of the biosensor would be to include an additional sample in which the body-fluid sample is spiked with a known concentration of the target compound. This sample could confirm if the measurement is affected by the matrix components. Furthermore, in a controlled environment, such as in the clinic, one could also use this biosensor to perform several additional control experiments and combine them with semi-quantitative measurements by quantifying the response using a calibration curve.

Comment 3.7: Could this system be used to detect human blood serum?

Response: To further demonstrate the capability of our biosensor to detect cannabinoids in different human samples, we performed **additional experiments** and confirmed that our system can determine THC present in reconstituted urine and blood serum samples. The new results are included in a separate panel in the revised **Figure 8g**.

Comment 3.8: THC is reported to have a strong psychoactive effect. Can the sensors developed in this work be used to distinguish structurally similar trans- Δ^9 -THC, CBD, CBC, and CBG, CBDA and CBGA? For drug screening or illegal THC detection, the detection of a full panel of CBD analogs will be more important. Please explain how to improve the specificity of the GPCR sensors.

Response: This is a very good suggestion. In response, we performed **additional experiments** to determine the ability of the biosensor to detect compounds that are structurally similar to THC or CBD. We selected to test: CBC, CBD, CBG, CBDA, CBGA and THCA. We found that CBC and THCA behave as agonists, whereas CBD, CBDA, CBG, CBGA act as antagonists. These results are now included in (**Figure 4f-k**).

The biosensor described in this work enables a number of different applications. We see that additional applications requiring high specificity towards one structural analog of a ligand relative to another could be unlocked in the future. Such biosensors could be developed by either engineering of the receptors using approaches, such as directed evolution (Sarkar et al., 2008 Proc Natl Acad Sci U S A 105, 14808-14813), or sourcing homologs with desired properties from other organisms. This has now been included in **lines 637-640** of the revised manuscript.

Reviewers' Comments:

Reviewer #2:

Remarks to the Author:

The authors have re-written portions of the manuscript and performed additional experiments that have addressed my criticisms to the extent that my criticisms could be realistically addressed.

Reviewer #3:

Remarks to the Author:

All my concerns have been addressed and please accept the current version of the manuscript.